



# Mediterranean Outflow Water variability during the Early Pleistocene climate transition

Stefanie Kaboth[1], Patrick Grunert[2], Lucas J. Lourens[1]

[1] Department of Earth Sciences, Faculty of Geosciences, Utrecht University, Heidelberglaan 2, 3584 CS, Utrecht, The Netherlands

[2] Institute of Earth Sciences, University of Graz, NAWI Graz, Heinrichstraße 26, 8010 Graz, Austria

*Correspondence to*: Stefanie Kaboth (S.Kaboth@uu.nl)

**Abstract.** Gaining insights into the evolution of Mediterranean Outflow Water (MOW) during the Early Pleistocene climate transition has been so far hampered by the lack of available paleoclimatic archives. Here we present the first benthic foraminifera stable oxygen and carbon isotope records and grain-size data from IODP Expedition 339 Site U1389 presently located within the upper core of the MOW in the Gulf of Cadiz for the time interval between 2.6 and 1.8 Ma. A comparison with an intermediate water mass record from the Mediterranean Sea strongly suggest an active MOW supplying Site U1389 on glacial-interglacial timescales during the Early Pleistocene. We also find indication that the increasing presence of MOW in the Gulf of Cadiz during the investigated time interval aligns with the progressive northward protrusion of Mediterranean sourced intermediate water masses into the North Atlantic, possibly modulating the intensification of the North Atlantic Meridional Overturning Circulation at the same time. Additionally, our results suggest that MOW flow strength was already governed by precession and semi-precession cyclicity during the Early Pleistocene against the background of glacial-interglacial variability dominated by the obliquity cycle of Earth`s inclination axis.

*Keywords: Mediterranean Outflow, Early Pleistocene, Atlantic Meridional Overturning Circulation, Sapropel*

## 1 Introduction

The Mediterranean Outflow Water (MOW) is a distinct hydrographic feature at intermediate water depths in the Gulf of Cadiz, distinguished from other ambient North Atlantic water masses by its warm and saline character (Ambar and Howe, 1979; Bryden et al., 1994; Bryden and Stommel, 1984). In the modern hydro-climatic setting of the Mediterranean Sea the MOW is predominately sourced by Levantine Intermediate Water (~70%), formed in the Eastern Mediterranean Basin, and variable parts of Western Mediterranean Deep Water (WMDW) originating in the Alboran and Tyrrhenian Sea (Millot, 2014, 2009; Millot et al., 2006). After exiting the Strait of Gibraltar, the MOW plume cascades down the continental slope due to its increased density (Ambar and Howe, 1979; Hernandez-Molina et al., 2014a; Hernández-Molina et al., 2006; Mulder et al., 2006). In the Gulf of Cadiz, MOW follows the topography of the continental shelf in two major flow cores at



800-1400 m water depth (lower MOW core), and 500-700 m water depth including our study area (upper MOW core, Fig.1)
(Baringer and Price, 1997; Borenäs et al., 2002; Hernández-Molina et al., 2013). After exiting the Gulf of Cadiz, most of
MOW flows north along the European continental margin until it mixes with the North Atlantic Current at Rockall Plateau
(Hernandez-Molina et al., 2014b).
Beyond the Mediterranean region, MOW has been acknowledged as an important modulator of the North Atlantic salt
budget with previous research suggesting that the absence of MOW may reduce Atlantic Meridional Overturning Circulation
(AMOC) by as much as 15% compared to modern (Rogerson et al., 2006). Despite its potential cosmopolitan significance
the paleoceanographic history of MOW has so far been only studied for the Pliocene (Khelifi et al., 2009; Khélifi et al.,
2014), and during the last climatic cycle (Bahr et al., 2015; Kaboth et al., 2016; Llave et al., 2006; Schönfeld, 2002;
Schönfeld and Zahn, 2000; Toucanne et al., 2007; Voelker et al., 2006). In this light, the reconstruction of MOW variability
might be particularly interesting in the broader view of the Pliocene-Pleistocene climate transition. The early Pleistocene
period spans the transition from the preceding Pliocene climate optimum with limited ice sheets in the Northern Hemisphere
to the cooler Middle and Late Pleistocene climate with rapidly developing continental ice growth in both hemispheres
(Raymo et al., 1992; Shackleton and Hall, 1984). Throughout the Early Pleistocene, however, an interruption of the long-
term Northern Hemisphere ice volume increase can be observed in concert with a sea-surface temperature stabilization in the
high latitude North Atlantic cooling trend (Bell et al., 2015). It was suggested that these changes relate to an increase in
AMOC strength, and in extension, an increase in northward heat transport (Bell et al., 2015).
Here we elaborate on the possible role of MOW on North Atlantic Paleoceanographic changes during the early Pleistocene
climate transition by investigating the benthic foraminifera stable oxygen and carbon   isotopes and grain-sizes from IODP
339 Site U1389, located on the upper slope of the Gulf of Cadiz (see Fig. 1) for two time intervals: 2.6 and 2.4 Ma and 2.1
and 1.8 Ma. We have compared our new data with benthic stable isotope record of the Singa/Vrica sections on Calabria
(Italy), representing the intermediate water mass endmember of the Mediterranean Sea (Lourens et al., 1996; unpublished
data) that serves as a reference for the source region of MOW during the Early Pleistocene (Fig. 1). Our results bridge the
gap in our understanding of MOW variability between the wider researched Pliocene and Late Pleistocene. We aim to shed
new light on MOW variability during the Early Pleistocene by analysing hydrographic changes within the Mediterranean
source region, investigating the low-latitude control of MOW against the background of dominant obliquity controlled
glacial-interglacial cyclicity and documenting the potential influence of MOW variability on long-term climatic oscillations
in the North Atlantic.

## 2 Material & Methods

### 2.1 Site U1389

Integrated Ocean Drilling Program (IODP) Site U1389 (36°25.515'N; 7°16.683'W) was drilled in December 2011 and
January 2012 during Expedition 339 (Stow et al., 2013). It is located on the southern Iberian Margin ~90 km west of the city



of Cadiz and perched on the northwest side of the Guadalquivir diapiric ridge in 644 m water depth (Fig. 1). At present,
IODP Site U1389 is directly influenced by the upper MOW core (Hernández-Molina et al., 2013; Stow et al., 2002, 2013).
For the present study we analysed 423 samples from Site U1389 Hole E which cover the Early Pleistocene (2.6 to 1.8 Ma)
time interval at 30 cm intervals between 549.8 to 706.35 mbsf. An expanded hiatus at Hole U1389E between 2.1 and 2.4 Ma
(~622-644 mbsf) has been initially related to a phase of highly active MOW (Hernández-Molina et al., 2013; Stow et al.,
2013). However, more recent findings link this compressional event to tectonically invoked erosion (Hernández-Molina et
al., 2015). As a consequence we present the data split in two intervals (Interval I: 2.6-2.4 Myr and II: 2.1 to 1.8 Myr).
**2.2 Stable isotope measurements and interspecies correction**
The freeze-dried sediment samples were wet sieved into three fractions (>150 μm, >63 μm and >38 μm), and their residues
oven dried at 40°C. Stable oxygen ($\delta^{18}$O) and carbon ($\delta^{13}$C) isotope analyses were carried out on 4 to 6 specimens of the
epifaunal living foraminiferal species *Planulina ariminensis* and *Cibicides ungerianus* from the >150 μm size fraction. All
selected specimens were crushed, sonicated in ethanol, and dried at 35°C. Stable isotope analyses were carried out on a
CARBO-KIEL automated carbonate preparation device linked to a Thermo-Finnigan MAT253 mass spectrometer at Utrecht
University. The precision of the measurements is ±0.08‰ for $\delta^{18}$O and ±0.03 for $\delta^{13}$C. The results were calibrated using the
international standard NBS-19, and the in-house standard NAXOS. Isotopic values are reported in standard delta notation ($\delta$)
relative to the Vienna Pee Dee Belemnite (VPDB).
*P. ariminensis* was absent in 100 samples; resulting gaps were filled with *C. ungerianus* values corrected for interspecies
isotopic offsets. The calculation of the interspecies offset is based on 62 paired isotope measurements of both benthic
species. The $\delta^{18}$O interspecies offset was determined by applying a least square linear regression equation (Fig. 2). The
Pearson correlation coefficient ($R2$) between both species shows high correlation of 0.79 for $\delta^{18}$O. The calculated slope of
this relationship is ~0.97 with an intercept of +0.10 ‰ between P. ariminensis and C. ungerianus.
In contrast, the $\delta^{13}$C correlation was insignificant with $R^2$ of 0.02 between the two benthic species (Fig. 2). Therefore, we
only present the $\delta^{13}$C of P. ariminensis, considered a valuable basis for δ13C studies of the paleo-hydrography of the MOW
(Zahn et al., 1987).
**2.3 Grain-size analyses**
The stable isotope sample preparation was used to obtain weight percentages (wt.-%) of the grain-size fractions >150 μm,
150-63μm, 63-38 μm and <38 μm for the investigated samples were obtained during sample preparation for isotope
analyses. We concentrate on the grain-size fraction between 63-150 μm which has been used previously as indicator for flow
strength changes in the Gulf of Cadiz attributed to MOW variability (Rogerson et al., 2005). Even though untreated weight
percentages hold a bias it has been shown for the last climatic cycle that weight percentages mirror major peaks in Zr/Al
records, considered a reliable recorder of MOW flow strength variability (Bahr et al., 2014), and thus can be used to trace
patterns of MOW flow strength variability (Kaboth et al., 2016).



## 2.4 Chronology

Primary age constraints are based on paleomagnetic and biostratigraphic tie points as listed in Table 1. The secondary age model follows the visual correlation of the benthic $\delta^{18}O$ record at Site U1389 to the benthic $\delta^{18}O$ "MedSea" stack of Lourens et al. (unpublished data) within the investigated time period. The MedSea stack is based on the benthic *C. ungerianus* $\delta^{18}O$ values from the Singa and Vrica sections located in Calabria, Italy derived from the same samples used for the planktic $\delta^{18}O$ record in Lourens et al., (1996a, pers. comm.). The stable isotope measurements for the MedSea stack were carried analogous the protocol described in section 2.2 (Lourens 2016, pers. comm.) The *C. ungerianus* values of the MedSea stack were adjusted to the *P. ariminensis* based $\delta^{18}O$ record at Site U1389 by applying the interspecies correction equation cited under section 2.2. The Mediterranean Sea stack $\delta^{18}O$ time series is based on tuning sapropel midpoints to La2004 65° N summer insolation maxima, including a 3-kyr time lag (Lourens, 2004). Monitoring of the sedimentation rate was done to control viability of secondary age model. The designation of MIS stages follows the MedSea stack chronology (Lourens, 2004). The respective tie points of the secondary age model are listed in Table 2.

## 2.5 Spectral Analysis

Spectral analysis was performed to test for statistically significant cycles with respect to orbital parameters. For analysis of orbital periodicities, the non-constantly sampled time series were analysed by a Multi Taper Method using the program REDFIT (Schulz and Mudelsee, 2002).

## 3 Results

### 3.1 Age model & Sedimentation rates

The two studied intervals of the Site U1389 δ18O record exhibit similar glacial-interglacial variability as present in MedSea stack throughout the Early Pleistocene. The estimated mean sedimentation rate for both intervals is ~0.30 m/kyr (Fig. 3) which is similar to the sedimentation rate of ~0.25 to ~0.30 m/kyr that has been calculated from shipboard stratigraphy for the past 3.2 Myr (Hernández-Molina et al., 2013; Stow et al., 2013). A doubling or tripling of the sedimentation rate coincides with transition of MIS 101 to MIS 100 and interglacials MIS 99 and MIS 97 in Interval I, and ~MIS 68 in Interval II. Condensed sections with low sedimentation rates of ~0.1 m/kyr correlate with the transition between MIS 98 to MIS 97 and MIS 95 in Interval I, and MIS 78 to MIS 75 in Interval II, respectively.

### 3.2 Stable oxygen and carbon isotopes

The comparison between both intervals of the $\delta^{18}O$ record at Site U1389 with the benthic $\delta^{18}O$ MedSea stack is shown in Figure 4. In Interval I, lightest values of 1.17 and 1.22 ‰ coincide with interglacials MIS 103 and 101, and the strongest glacial enrichment in $\delta^{18}O$ (2.69 ‰) coincides with MIS 100. Transitional depletion is on average 0.97 ‰ with highest





values (1.29 ‰) in the interval between MIS 101 and 100 (see Fig. 4). In Interval II, the lightest values coincide with MIS 73
(1.36 ‰) whereas the strongest glacial $\delta^{18}$O enrichment can be observed during MIS 78, 72 and 68 with 2.47 ‰, 2.42 ‰ and
2.69 ‰, respectively (see Fig. 4). Transitional depletion is on average 0.82 ‰ with highest values (1.06 ‰ and 1,19 ‰) in
the interval between MIS 73 and 72, and the transition from MIS 69 to MIS 68. Pronounced amplitude offsets between the
$\delta^{18}$O signal of Site U1389 and MedSea are visible in both intervals but especially during MIS 103, 102, 77, 75 and 67 (Fig.
4). These perturbations are of the order of up to ~0.5 ‰ (e.g. MIS 75).
The comparison between both intervals of the $\delta^{13}$C record at Site U1389 with the $\delta^{13}$C MedSea stack is shown in Figure 4.
During Interval I, lightest values of 0.27 and 0.32 ‰ coincide with MIS 101 and 100, and the heaviest values (~1.27 ‰)
coincide with the transition of MIS 102 to MIS 101, MIS 100, and the transition between MIS 99 to MIS 98. In Interval II,
the lightest values correspond to MIS 74 (-0.02 ‰) and the transition between MIS 68 and 67 (-0.06 ‰). The heaviest $\delta^{13}$C
values coincide with MIS 71 (1.56 ‰).
**3.3 Grain-size**
The mean grain-size values (63-150 μm) for both investigated intervals are ~8.0 %-wt. Highest values of both investigated
intervals of up to ~60 %-wt. are correlated with MIS 100 and 77 (Fig. 4). The grain-size variability is seemingly not related
to glacial-interglacial variability as a clear response of the grain-size to the variability .of δ18O records at Site U1389 cannot
be observed.
**3.4 Spectral analyses**
The grain-size records of Interval I and II at Site U1389 exhibit significance (80% to 90%) variance in the precession (~23
kyr) and semi-precession (~ 11 kyr) frequency band (Fig. 5). The obliquity signal is insignificant in both investigated
intervals.
**4 Discussion**
**4.1 Glacial-Interglacial MOW variability at Site U1389 during the Early Pleistocene**
Site U1389 reveals $\delta^{18}$O values similar to the MedSea stack during Interval I (2.6-2.4 Ma) and Interval II (2.1-1.8 Ma),
which emphasize the direct influence of intermediate Mediterranean water masses on MOW (Fig. 4). This suggests that
MOW formation during the Early Pleistocene was similar to modern conditions where MOW originates largely from
intermediate water masses such as the Levantine Intermediate Water (Millot, 2009, 2014; Millot et al., 2006).
The $\delta^{18}$O difference between Site U1389 and the Mediterranean Sea is small during glacial periods in both investigated
intervals, suggesting that Site U1389 bathed in MOW during these colder climatic conditions throughout the early
Pleistocene time interval (Fig. 4). This is particularly interesting in light of the proposed vertical shift of the MOW flow path
during glacial periods of the Late Pleistocene fostered by the increased density of the outflowing Mediterranean water



masses (Kaboth et al., 2015; Lofi et al., 2015; Toucanne et al., 2007b; Voelker et al., 2006; Rogerson et al., 2005; Schönfeld
and Zahn, 2002). This suggests that Site U1389 was not subjected to major glacial-interglacial induced flow path changes
during the early Pleistocene, possibly due to its deeper and relatively proximal location to the Strait of Gibraltar, placing it
more into the general flow path of upper MOW. These results confirm the inferences derived from Site U1389 of the Late
Pleistocene interval where MOW activity was also shown to be largely unaffected by glacial-interglacial variability but
instead predominately influenced by insolation driven hydro-climatic changes of its Mediterranean source region (Bahr et
al., 2015).
In contrast, the interglacial periods of both intervals show a small but relative depletion in the Mediterranean Sea compared
to the $\delta^{18}$O signal at Site U1389 which might reflect relatively higher temperatures or lower salinity of the intermediate
Mediterranean Sea waters with respect to the MOWs during interglacial periods. The strongest intervals of relative $\delta^{18}$O
depletion throughout both investigated time periods correlate with MIS 103, 102, MIS 75 and MIS 67 characterized by a
depletion of up to ~0.5 ‰ in the Mediterranean Sea compared to Site U1389. This shift might correspond to a freshening of
the Mediterranean Sea intermediate water column during sapropel formation and a consequently reduction of MOW
influence at Site U1389 (Rogerson et al., 2012). In case of MIS 102 and 67 sapropels have been documented in the Eastern
Mediterranean Sea basin but not for MIS 75 (Emeis et al., 2000; Lourens, 2004; Lourens et al., 1992, 1996a).
During Interval II, the generally heavier $\delta^{13}$C values at U1389 are close to those of the Mediterranean Sea values inferring
that MOW was in fact the predominant source of bottom water at Site U1389 between 1.8 and 2.1 Ma. In contrast, the older
Interval I is characterized by a slightly increased $\delta^{13}$C gradient between Site U1389 and the Mediterranean Sea suggesting a
generally larger contribution of ambient North Atlantic water masses carrying a lighter $\delta^{13}$C signal to the site. This could
indicate a more vigorous MOW or that during Interval I the MOW flow core was less proximal than during Interval II. The
later argument seems to be supported by the grain-size and its variability, as Interval II shows a ~10% decrease in mean and
amplitude relative to Interval I (Fig. 4). This would suggest that during Interval I Site U1389 was less proximal to the flow
core albeit more sensitive to flow strength changes whereas during Interval II the MOW plume has settled upon Site U1389.
This is further supported by findings from seismic records in the Gulf of Cadiz that also suggest that at ~2.1 Ma the present
day circulation established (Hernandez-Molina et al., 2014b)
A distinct increase in the $\delta^{13}$C gradient can be seen during MIS 96, which may document a particular strong MOW activity.
However, the sample resolution during MIS 96 and the subsequent MIS 95 is relatively low so that increase in the $\delta^{13}$C
gradient remains ambiguous. The onset of the subsequent hiatus which has been argued to represent depositional erosion due
to increased bottom current activity of the MOW could argue for a strong intensification of MOW activity (Hernandez-
Molina et al., 2014b).





## 4.2 Precession control on MOW strength during the Early Pleistocene: Similarities to Late Pleistocene MOW behaviour?

Untreated grain-size weight percentages can only give an indication for patterns in flow strength (Kaboth et al., 2016). For the two investigated intervals we find that the 63-150 µm fraction variability is seemingly modulated by a ~23 kyr pacing (Fig. 4). This relationship is evident in the power spectrum of the grain-size data which yields for both intervals a dominance in the precession frequency band (~23 kyr); more prominently in the younger than in the older interval (Fig. 5). This suggests that the flow strength of MOW was probably directly modulated by precession during the Early Pleistocene, aligning with previous findings based on Zr/Al ratios at Site U1389 from the Late Pleistocene (Bahr et al., 2015). For the late Pleistocene, an inverse relationship was found between precession and MOW dynamics (Bahr et al., 2015; Kaboth et al., 2016). During periods of increased summer insolation at the time of precession minima, the monsoonal rain belts expand northward causing an increase of freshwater discharge by the river Nile (e.g. Rohling et al., 2015; Rossignol-Strick, 1985, 1983). This effectively impedes intermediate water mass formation in the Eastern Mediterranean, thereby suppressing MOW production. From the correlation of the filtered ~23 kyr signal to the grain-size variability at site U1389 a similar relationship already existed during both investigated intervals of the Early Pleistocene (Fig. 4). We also find significant semi-precession (~11 kyr) influence indicative for a primarily low-latitude response argued to originate in the tropics (Rutherford and D'Hondt, 2000; de Winter et al., 2014).

The $\delta18O$ signal comparison of Site U1389 and the MedSea stack is also particular interesting in the context of sapropel formation, as the MedSea stack due to its intermediate paleo-water depth was sensitive to freshwater induced changes in the intermediate water composition. A substantial freshening of the intermediate water masses in the Mediterranean Sea can be inferred from the strongly depleted $\delta^{18}O$ values during MIS 103, 102, 77, 75 and 67 relative to Site U1389 (Fig. 4). The potentially reduced MOW supply at Site U1389 at the same time would increase the isotopic gradient between both locations, as Site U1389 could be affected by more open ocean conditions. However, despite the low sample resolution, this seems not a persistent relationship throughout both investigated intervals. For the Holocene S1, the proposed reduction in MOW has been documented by the absence of sandy contourite layers from the middle slope of the Gulf of Cadiz indicating a sudden reduction in flow strength and sediment delivery by the MOW (Toucanne et al., 2007; Voelker et al., 2006). The grain-size values throughout both investigated intervals at Site U1389 are typically low during sapropel formation supporting the findings from the middle and upper slope during the Late Pleistocene (Kaboth et al., 2016). However, the grain-size is seemingly increased during the sapropels deposited in the Eastern Mediterranean Sea at ~1.92 and 1.85 Myrs (Fig. 4). This in-phase behaviour could potentially be a tuning artefact or relate to the fact that numerical model simulations imply that remnant thermal riven overturning circulation still occurs throughout the most extreme freshening events in the eastern Mediterranean Sea (Myers, 2002). This would imply that during the sapropel formation at ~1.92 and 1.85 Myrs MOW was potentially still active at Site U1389.





**4.3 Did MOW contribute to the Early Pleistocene climate transition?**

Between ~2.8 and 2.4 Myrs (Interval I) occurrences of *Neogloboquadrina atlantica* (sin), an extinct polar species, were reported in the Mediterranean Sea during glacial periods suggesting the intrusion of colder water masses into the Mediterranean basin (Becker et al., 2005; Lourens and Hilgen, 1997; Zachariasse et al., 1990). We also find *N. atlantica* (sin) present during glacial periods of Interval I (Fig. 4), confirming a more southern delineation of transitional and subpolar water masses during glacial periods of the Early Pleistocene than in recent setting (Voelker et al., 2015). This latitudinal shift might occurred in concert with a more sluggish AMOC at least during the glacial periods if not throughout the whole time interval (Bell et al., 2015). Colder and more arid background conditions in the Mediterranean Sea could foster a stronger MOW analogous to cold spells related to Heinrich Events throughout the last climatic cycle (Bahr et al., 2014, 2015; Kaboth et al., 2016). An intensification of MOW during Interval I would align with the increased $\delta^{13}$C gradient between Site U1389 and the Mediterranean Sea suggesting a more vigorous MOW which is also reflected by higher grain-size amplitudes compared to Interval II (Fig. 4). Our data, however, do not extend further back in time to test whether these conditions coincides with the proposed steady increase of MOW activity in the Gulf of Cadiz since 3.2 Ma as inferred from natural gamma ray logs and seismic profiles (Hernández-Molina et al., 2015), and with the arrival of Mediterranean sourced intermediate water mass at North Atlantic Sites DSDP 548 and 552 and ODP 982 from ~3.6 onwards (Khélifi et al., 2014; Loubere, 1987). This northward protrusion of warm and saline MOW towards high-latitude deep-water convection hot spots is considered an important modulator of the North Atlantic salt budget (Bahr et al., 2015; Rogerson et al., 2006; Voelker et al., 2006). We suggest that steady contributions of MOW throughout Interval I supplied continuously salt into the North Atlantic and potentially preconditioned the strong AMOC activity phase starting at ~2.4 Ma (Bell et al., 2015) when a tipping point was reached. The Early Pleistocene MOW might therefore have acted as a positive climatic feedback mechanism against the background of increasingly colder temperatures (Fig. 4). This stands in contrast to the warm Pliocene setting where it was proposed that MOW contributions to the North Atlantic did not have a significant influence on the AMOC (Khélifi et al., 2014).

The intensification of the AMOC is also in concert with the disappearance of *N. atlantica* (sin) in the Mediterranean Sea and the North Atlantic up to at least 52°N after ~2.4 Ma (Lourens and Hilgen, 1997; Weaver and Clement, 1987). This suggests the reduction in southward protrusion of colder water masses and hence the N. atlantica extinction, and a return to a warmer background climate in the Mediterranean region during glacial periods (Lourens, 2008).

The increased AMOC activity is documented by the North Atlantic SST record of Site ODP 982 displaying a plateau starting at ~2.4 Ma indicating more steady climate conditions (Fig. 4), and a stagnation in Northern Hemisphere ice sheet growth (Bell et al., 2015; Lawrence et al., 2009). Coinciding with this stabilization of North Atlantic SSTs is a cooling in the South Atlantic attributed to a northward piracy of the tropical warmer water pool by a strong AMOC and implying an active interhemispheric climatic seesaw at that time (Patterson et al., 2014; Etourneau et al., 2010). Despite the lack of direct data at Site U1389 between the 2.4 to 2.1 Ma interval, seismic records from the Gulf of Cadiz suggest that the hiatus represents a

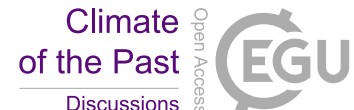

depositional erosion feature caused by intensified bottom current activity, and hence strong MOW flow (Hernandez-Molina
et al., 2014b). This would align with the continuous strong AMOC activity in the North Atlantic (Bell et al., 2015).
From the reduction of the $\delta^{18}$O and $\delta^{13}$C gradient between Site U1389 and the MedSea stack (Fig. 4), it appears that after
~2.1 Ma MOW settled and upon Site U1389 (Fig. 4). The reduction in grain-size might also imply more stable MOW
behaviour whereas during the transitional phase of the older Interval I MOW was probably more erratic, indicated by the
high grain-size variability and the increased $\delta^{13}$C gradient (Fig. 4). Unfortunately, we lack data beyond ~ 2.5 Ma from ODP
Sites 549, 552 and 982 to further trace the temporal MOW influence in the high-latitude North Atlantic throughout Interval
II but it stands to reason that continued MOW contributions also during Interval II might have contributed to the sustained
AMOC activity.
**5 Conclusions**
Based on our results, the supply of MOW to Site U1389 was already established during the Early Pleistocene and not limited
to Late Pleistocene climate conditions. In addition, we find indication that the MOW flow strength might have been
modulated by precession superimposed on glacial-interglacial change, this aligns with findings from the Late Pleistocene at
Site U1389 and suggests that Site U1389 is a true recorder of MOW variability also throughout Early Pleistocene. In the
broader view of the Early Pleistocene climate transition we find indication that increased MOW might have contributed to
the increased AMOC phases starting from 2.4 Ma, and thus influencing North Atlantic oceanic heat transport.
**Data availability**
The data related to this manuscript is already stored in PANGEA and placed under moratorium. After a final decision has
been made about the publication of this manuscript the data will made publicly accessible.
**Author contribution**
S. Kaboth conducted the measurements, data analyses, and prepared the manuscript with contributions from P. Grunert and
L. Lourens. Additionally, L. Lourens and P. Grunert provide financial support to this submission.
**Competing interests**
The authors declare that they have no conflict of interest.





## Disclaimer


Figure 1 (Study area) is a modified version of an already published figure (Hernández-Molina et al., 2013; Stow et al., 2013).
The figure is used under creative commons Attribution 3.0 unported licence.

## Acknowledgements


We acknowledge the Integrated Ocean Drilling Program (IODP) for providing the samples used in this study as well as A.
van Dijk at Utrecht University for analytical support. This research was funded by NWO-ALW grant (project number
865.10.001) to Lucas J. Lourens and contributions from project P25831-N29 of the Austrian Science Fund (FWF).

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







**Figure 1: Study area with illustration of modern MOW pathways modified after (Hernández-Molina et al., 2013; Stow et al., 2013). Site location of U1389 (red dot) and the location of the Singa/Vrica section in Italy (black dot) are marked.**



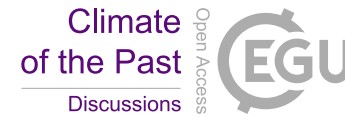


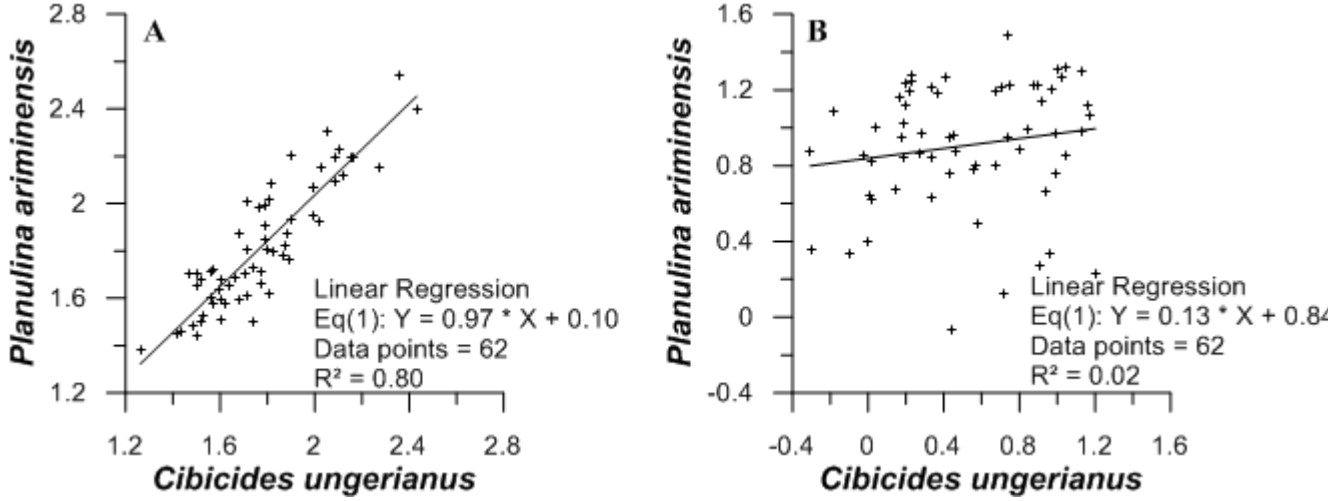


**Figure 2: The δ¹⁸O and δ¹³C interspecies correlation between benthic foraminifera *Cibicides ungerianus* and *Planulina ariminensis***
**at Site U1389. Parallel measurements were conducted throughout both investigated intervals. Linear square regression (red line)**
**equation and Pearson correlation coefficient ($R^2$) are shown.**

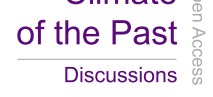

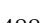


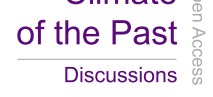


**Figure 3: Chronology of Site U1389. Assigned marine isotope stages (MIS) follow Lourens et al. (2004). (A) Both intervals of the δ18O record of Site U1389 on shipboard MCD scale correlated to the benthic δ$^{18}$O record of the Mediterranean Sea (MedSea stack) after Lourens et al. (1996, unpublished data). Chronostratigraphy of MedSea stack is based on tuning sapropel midpoints to La2004 65° N summer insolation (Lourens, 2004). Lines with arrows indicate selected tie points used for the age model (a full list of tie points is available in Table 2). Black triangles with numbers indicating used biostratigraphic and paleomagnetic tie points as referenced in Table 1. Black and white bar at the top represents core recovery following Hernández-Molina et al. (2013) (B) Comparison of the benthic δ$^{18}$O record of Site U1389 on new time scale according to our tuning, and the benthic δ$^{18}$O MedSea stack on its respective age model (Lourens et al. 2004) (C) Calculated sedimentation rates for Site U1389. Mean Sedimentation rate is marked by dotted line.**





499

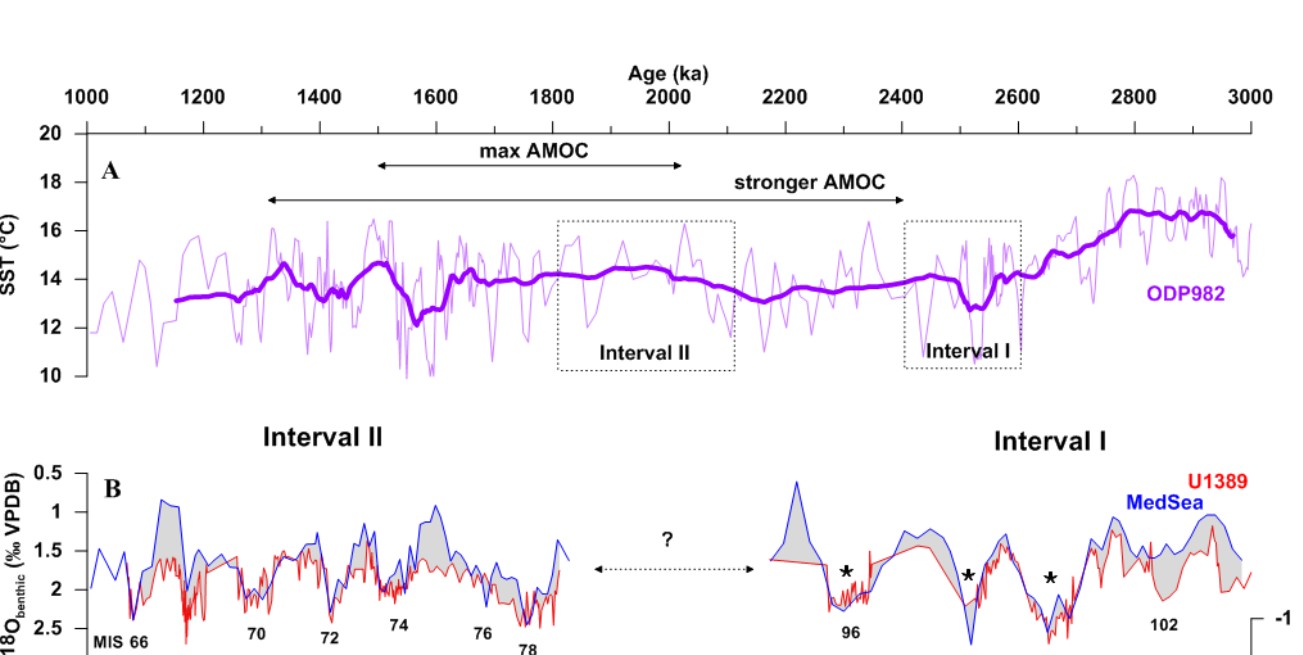

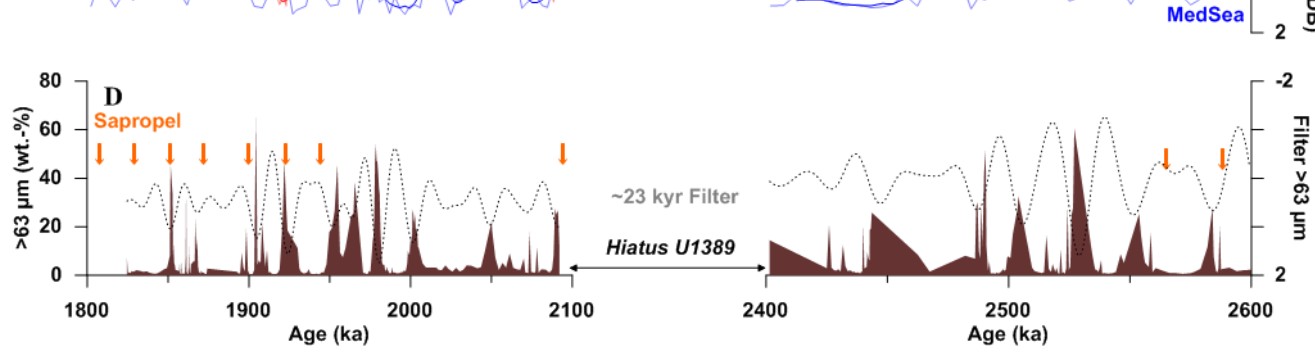

500

**Figure 4: (A) UK37 based sea-surface temperature (SST) record of North Atlantic Site ODP 982 (Lawrence et al., 2009). The running mean has a band width of 23. AMOC phases are marked by black arrows and follow the chronology of Bell et al. (2015). (B) Benthic δ18O records of both investigated intervals at Site U1389. Interval I comprises the time frame of 2.6 to 2.4 Ma and Interval II 2.1 to 1.8 Ma. Isotopic gradient between both records is indicated by the grey-shaded area. (C) Comparison of δ13C of P. ariminensis for both investigated intervals at Site U1389 and δ13C of the MedSea stack (Lourens et al. 1996, unpublished data). The running means have a band width of 5. The C. ungerianus based δ13C values of the MedSea stack were adjusted to P. ariminensis δ13C values of Site U1389 following the interspecies correction presented in Kaboth et al., (in prep) (D) Grain-size (63-150 μm wt.-%) records for both investigated intervals at Site U1389. The filtered ~23 kyr signal of the grain-size signal is indicated by the black dotted-line. Sapropel mid-points are marked by orange arrows and follow the chronology of Emeis et al. (2000).**




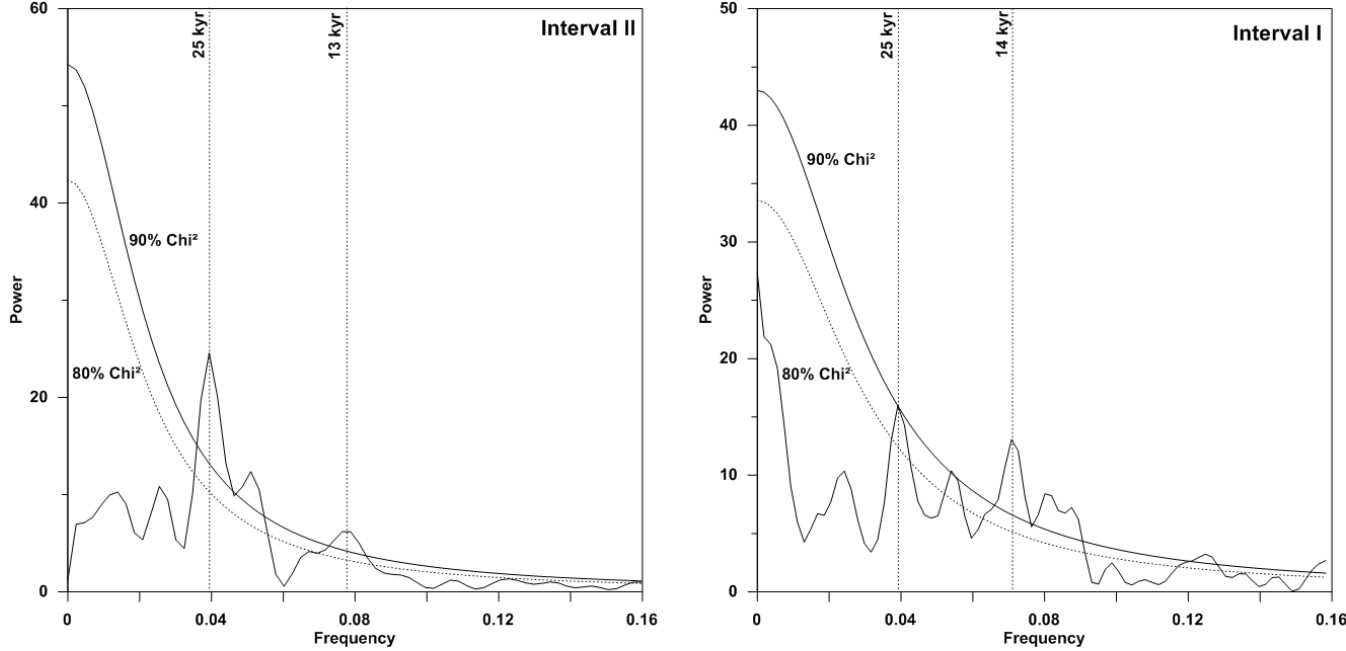


**Figure 5: REDFIT Power Spectra of the grain-size values (63-150µm fraction in wt.-%) for both investigated intervals of Site U1389 (Interval I = 2.6-2.4 Ma ; Interval II: 2.1-1.8 Ma). The 90% (red) and 80% (blue) confidence levels are given.**




| No. | Event | TOP Depth (mbsf) | BOT Depth (mbsf) | Time (Ma) | Reference |
|-----|-------|------------------|------------------|-----------|-----------|
| 1 | Top Olduvai | 542.00 | | 1.806 | 1 |
| 2 | Bottom Olduvai | | 592.00 | 1.945 | 1 |
| 3 | Matuyama/Gauss | 696.00 | | 2.588 | 1 |
| 4 | LO *C. macintyrie* | 510.09 | 515.65 | 1.66 | 2 |
| 5 | FO *G. inflata* | 627.21 | 630.21 | 2.09 | 3 |
| 6 | LO *G. puncticulata* | 645.02 | 646.61 | 2.41 | 3 |
| 7 | LO *D. pentradiatus* | 674.25 | 681.98 | 2.5 | 2 |
| 8 | LO *D. scurlus* | 681.98 | 693.70 | 2.53 | 2 |
| 9 | LO *D. tamalis* | ~805 | | 2.8-2.87 | 4 |


**Table 1: Paleomagnetic and biostratigraphic tie points used in the primary age model of Site U1389 based on shipboard data following Hernández-Molina et al. (2013) and Stow et al. (2013). 1 = Gradstein et al. (2012); 2 = Raffi et al. (2006); 3= Lourens et al. (2004); 4 = Grunert P., pers. comm.**






| Depth (mbsf) | Age (ka) |
|---|---|
| 512 | 1660 |
| 542 | 1806 |
| 551.25 | 1828 |
| 554 | 1851 |
| 564 | 1861 |
| 570 | 1867 |
| 574 | 1875 |
| 580 | 1898 |
| 592.00 | 1945 |
| 595.00 | 1965 |
| 600 | 1975 |
| 615.63 | 2005 |
| 623.00 | 2070 |
| 629.1 | 2092 |
| 629.75 | 2117.5 |
| 631.1 | 2132.5 |
| 646 | 2425 |
| 648.75 | 2435.5 |
| 665.1 | 2462.5 |
| 666.3 | 2486 |
| 673 | 2500 |
| 677.45 | 2517.5 |
| 687 | 2539 |
| 689 | 2552 |
| 691.5 | 2560 |
| 693.5 | 2583 |
| 696 | 2588 |
| 805 | 2800 |

**Table 2: Paleomagnetic and biostratigraphic tie points used in the primary age model of Site U1389 based on shipboard data**
**following Hernández-Molina et al. (2013) and Stow et al. (2013). 1 = Gradstein et al. (2012); 2 = Raffi et al. (2006); 3= Lourens et**
**al. (2004); 4 = Grunert P., pers. comm**