# Peer review of "Mediterranean Outflow Water variability during the Early Pleistocene"

_Climate of the Past, 2017_

## Referee Comment (RC1) · Anonymous Referee #1 · 19 Mar 2017

Dear Climate of the Past Editorial Board

I hereby you receive my report on the MS " Mediterranean Outflow Water variability during the Early Pleistocene climate transition" by Kaboth et al.

The authors provided new important information on Mediterranean Outflow Water (MOW) during the Early Pleistocene comparing benthic stable isotopic data from Mediterranean and Atlantic Ocean records. In particular, the authors describe the change of MOW in two time intervals, between 2.6 and 2.4 and between 2.1 and 1.8 Ma. Notwithstanding the different time resolution of the two compared study records, the comparison of oxygen isotope signals is clear and suggest new hypothesis concerning the oscillation of Mediterranean Outflow Water in the Gulf of Cadiz and possible modulation of the North Atlantic Meridional Overturning Circulation.

[Figure]

The manuscript is properly constructed and it is evident that the data support the interpretation proposed in the manuscript. Specific comment: Chapter Age model & Sedimentation rates It is clear evident in Fig 3 that the U1389 marine record shows a variable Sed Rate, so that it is not possible to use the term mean Sed Rate. It has no sense. Maybe it is important to associate this change in Sed Rate to particular events.

Chapter Spectral analyses I am convince that the age model is correct, but I am not convince that is necessary the spectral analysis in the paper. In addition, the possible information concerning the astronomical periodicities are marginally used in the interpretation and discussion of data. If the authors want to really used the spectral analysis I think that it is necessary to go in detail with wavelet analysis and detail comparison with astronomical insolation curve. This further analysis will improve the manuscript.

Chapter Glacial-Interglacial MOW variability at Site U1389 during the Early Pleistocene The discussion concerning glacial and interglacial period seems to be robust. I am not convince about the MIS102. The comparison between oxygen isotopes is not as clear as the other intervals. In my opinion, the authors need to support and render more clear this correlation and explain the difference. My overall comment concerning this chapter is that the authors have to describe in detail the comparison between the two study records. The authors use also the grain-size oscillation to support the different power of MOW in two study intervals. In my opinion, this proxy is weak and the % differences suggested by the authors are very weak. The authors suggest the d13C gradient between the Mediterranean and Atlantic site as proxy for different contribution to North Atlantic water masses. I think that the authors need to explain this correlation and interpretation.

Chapter Precession control on MOW strength during the Early Pleistocene: Similarities to Late Pleistocene MOW behaviour? I think that it is necessary to add in Figure 4 the astronomical parameter of precession and I think that the authors need to run wavelet analysis on proxy record of grain-size to understand the stability of precession frequency band (23 ky). Also in this chapter, the authors use the grain-size suggesting

an increase during the sapropel deposition in the Eastern Mediterranean. This datum is weak and it is evident only in two time intervals.

Chapter Did MOW contribute to the Early Pleistocene climate transition? In this chapter, the authors introduce the Neogloboquadrina atlantica left coiled and its value to monitor the southern delineation of transitional and subpolar water masses during glacial period. However, where is the distribution of this species? I think that the authors need to plot the abundance pattern of Neogloboquadrina atlantica left coiled for Mediterranean and Atlantic records. In addition, it is important to document the correlation between this species and the precession curve. The authors suggest a correlation between SST of North Atlantic South Atlantic. Probably it could be useful to propose a figure with a comparison of the SST of north and south Atlantic.

Figure: Fig. 1 – Due the issue of the manuscript I think that it is important to add a figure of the Mediterranean water masses circulation. Fig. 3 – In the upper part of this figure the authors show a correlation between U1389 site and Mediterranean record. However, for the Interval I, it is not clear the position of the third correlation lines for MIS 101. I think that is only a graphic mistake. Please try to check. Fig. 4 – It is necessary to add the precession curve, to add the arrows of the sapropels also in the part B of the figure. The ages in the manuscript are reported in Ma while in this figure in ka. Please try to use the same scale. Concerning the grain-size curve, I do not think that is correct to plot a continuous curve, mainly for the Interval I, where the sample are far from each other, (between 2.4 and 2.5 Ma). Try to use another graphic representation. Fig. 5 – Please add the red noise level in the spectra.

Table: In table 1 it is necessary to write the name of the calcareous plankton species (i.e., Globorotalia puncticulata). Be careful, Raffi et al 2006 is not reported in the reference In table 2, Raffi et al 2006 is not reported in the reference

The manuscript results properly constructed and all figures are representative and useful for this version of the manuscript.

I think that it is very important to publish these data, because of in the scientific literature there are very few papers suggesting a correlation between Atlantic and Mediterranean during the Early Pleistocene. In addition, this correlation suggests new important evidences for the contribution of MOW to Atlantic circulation. My overall conclusion is that the paper is suitable for the journal but unfortunately it needs still moderate to minor revision concerning the presentation of data and power spectra analysis. In addition, I think that a detailed correlation with oscillation of astronomical parameter is necessary. In several parts of the manuscript, the authors used the grain-size proxy, but the differences reported by the authors are too weak to support the differences of MOW in the two study time interval.

---

## Referee Comment (RC2) · Anonymous Referee #2 · 9 May 2017

The manuscript " Mediterranean Outflow Water variability during the Early Pleistocene climate transition" by Kaboth and others is a nice contribution to already existing paleoceanographic reconstructions focusing on the MOW. In particular, the authors apply stable isotopes on benthic foraminifera and grain size analyses to determine MOW variability and its impact on the Atlantic Meridional Overturning Circulation (AMOC) during the early Pleistocene. It is nicely written, easy understandable and fits into the scope of Climate of the Past. However, as I have a few comments this MS should be considered for publications after moderate revisions. Please find below my comments:

Major comments:

1) The question how the MOW influenced AMOC is still not understood and needs further investigation and consideration. The present dataset may shed some light into

this debate. However, I would suggest that the authors not only compare their data to the SST reconstruction by Lawrence et al. but also to other records (e.g. Naafs et al. 2010, Khélifi & Frank 2014, Lisiecki 2014)? I think to make such a statement the authors should compare more thoroughly their records to other published records.

2) The authors use two different foraminifera Planulina ariminensis and Cibicides ungerianus, which are considered to be both epibenthic species and thus record ambient seawater characteristics. I wonder why the carbon isotopic composition of P. ariminensis shows a smaller spread than C. ungerianus? There are studies that show similar values for P.ariminensis and the Cibicides group. Maybe the authors could highlight the samples of cold and warm periods differently in the x/y plot? Is that a respiration change of the foraminifera? Is there a change in $\Delta\delta$13C gradients between these two species? To solve this problem is of great importance for subsequent studies using these species, especially in the Gulf of Cadiz. Furthermore, the authors decided to restrict their $\delta$13C record to P. ariminensis. They mention a paper in prep by Kaboth including a interspecies-correction and an unpublished dataset of Lourens et al. and refer to this as an end-member for their recorded signal. However, this record is based on C. ungerianus $\delta$13C values. How does that work? Can the authors please clarify?

3) This study shows that MOW variability was probably paced by precession and semi-precession cyclicity during the early Pleistocene. In fact this was already shown for records of the eastern Mediterranean Sea (ODP site 967 and 969) and have been linked to North Atlantic records during a similar time period (MIS 100, Becker et al.). I think the authors should include this in their discussion.

4) Vertical movements of the different branches of MOW have been discussed in detail in many studies for the late Pleistocene (e.g Schönfeld et al.; Zahn et al.). To my opinion the authors should add more discussion about vertical movements of the MOW in the Gulf of Cadiz. Is it really possible to disentangle these different effects with $\delta$18O gradients? It is not really clear how you can make this statement from $\delta$18O values? Do you infer density from that according to Lynch-Stieglitz et al. 1999? This section

needs some more explanations.

Minor comments in the text:

Title: I am not aware of any Early Pleistocene climate transition. It is either the Mid-Pleistocene-Transition (1.25 to 0.7 Ma, Elderfield et al. 2012) or the Plio/Pleistocene transition (boundary). Major oceanographic changes during the early Pleistocene occur around 1.7 Ma (e.g Khélifi & Frank 2014, Hodell & Venz-Curtis 2006; Lisiecki 2014, Martinez Garcia et al., 2010, Ravelo et al., 2004). Therefore, I think the authors should consider to change the title.

Line 51: Can you please give more information about the reference site Singa/Vrica?

Line 112-118: How do you explain the large differences in sedimentation rates varying from 0.1m/kyr to 1.2m/kyr?

Line 216: You discuss the abundance of the extinct planktonic foraminifera N.atlantica (polar species) during cold periods? So I guess the black stars in Interval I (Figure 4) are the occurrences of. N. atlantica?

Line 263: The hiatus extends from 2.4 to 2.1 Ma, so how can you state that AMOC increased at 2.4 Ma from you data?

Please check throughout the text and figure captions for the superscription in $\delta$13C and $\delta$18O.

Figures: Figure 1: A detailed map of the study location is always necessary, however, in this case an additional map showing the exact location of MOW in the water column would be very helpful to fully understand the MOW flow in the Gulf of Cadiz.

Figure 2: This figure indicates the (non)correlation of the oxygen and carbon isotopic compositions of the two different analysed benthic foraminifera. However, this figure needs some improvement. As mentioned above the different stratigraphic/climatic intervals should be highlighted. Furthermore, it is not clear which figure (a or b) shows

carbon or oxygen. The reader can only guess from the data.

Figure 3: Replace commas with dots

Figure 4: This figure is confusing as it intuitively suggests that the Lawrence record covers the similar interval investigated in this study. Please change accordingly and add also other records.

---

## Author Comment (AC1) · 8 Jun 2017

Dear Editor,

Firstly, we would like to thank our two anonymous Referees for their careful handling of the manuscript as well as for the constructive comments and suggestions which we kindly acknowledge. The main criticism of the two referees generally concentrated on: (1) the validity of the grain-size distribution and its corresponding spectral analysis (Referee 1), (2) the applied benthic stable isotope correlation (Referee 2), and (3) the presentation of the data in Figure 4 (Referee 1 +2). In the following, we would like to give a detailed explanation concerning the comments and recommendations given by the Reviewers, as well as the intended revisions in our revised manuscript.

[Figure]

Stefanie Kaboth, on behalf of all co-authors

Referee 1

1.) Chapter Spectral Analysis & Precession control on MOW strength during the Early Pleistocene The spectral analysis will be improved as requested by Referee 1, and the corresponding section 4.2 in the manuscript will be enhanced to properly reflect changes in the stability of the precession signal throughout our investigated intervals and its relation to insolation at 30°N. For this, we will include the insolation variability at 30°N to Figure 4A, and also add wavelets analysis for both investigated intervals which will be added to Figure 5 as Figures 5C and 5D. These adaptions are also in line with suggestions made by Referee 2 (see below) to include the findings of Becker et al. (2006, 2005) about precession related signals in the Eastern Mediterranean Sea during interval I. The methodological description of the wavelet analysis will be added to Method section 2.7 (section 2.6 in the initial manuscript).

2.) Chapter Glacial-Interglacial change We believe that the MIS102 interval is robust in its represented version and no adaptions will be necessary. Following the age constrain of MIS 102 according to Lisiecki and Raymo, (2005) the interval extends from 2.575 to 2.554 Ma. Through the initial age model based on bio- and magnetostratigraphy (see Table 1 in our initial manuscript) the age interval between 2.5 and 2.588 Ma is very well constrained. In contrast to Referee 1, we argue that the grain-size variability is a reliable representation for MOW variability during our investigated intervals despite in-part low recovery and sample resolution. As stated in the initial manuscript (see Lines 90 to 93) the grain-size proxy has been successfully used for Late Pleistocene studies (i.e. Kaboth et al. 2016). We think that the argued absolute amplitude reduction in MOW flow strength between interval I and II (see Lines 173 in the initial manuscript) is also visible in Figure 4D and does not relay solely on the calculated relative change in amplitude expressed in Line 173. We also believe that the description of the change in $\delta$13C through North Atlantic water influence is sufficient in its current state (see Lines 168 to 177).

3.) Chapter: Did Mow contribute to the Early Pleistocene climate transition? The occurrence of Neogloboquadrina atlantica (sin) is denoted in Figure 4 with black stars during mid-MIS 100, 98, 96 at Site U1389. However, as also stated by Referee2 this occurrence pattern is not clearly understandable for readers in its initial version, and Figure 4 will be modified to this respect in the revised manuscript. The reference for abundance patterns of N. atlantica in the eastern Mediterranean Sea and North Atlantic are already provided in the discussion section 4.3 (Lines 239). The south Atlantic SST record of Site ODP 1090 will be added to Figure 4A following Bell et al. (2015) (this also aligns with suggestions made by Referee 2, see below).

Figures 1.) Figure 1: A sub-figure (Figure 1B) highlighting the Mediterranean Sea circulation will be added. 2.) Figure 3: Indeed, the correlation arrow for MIS 101 in Figure 3A is graphically displaced. This will be revised. The dotted line indicating the mean sedimentation rate in Figure 3C will be removed. 3.) Figure 4: In the revised version we will unify the age notations between Figure 4 and Table 1. As stated above the insolation curve will be added to the revised Figure 4. However, the mid-point levels of sapropels throughout the investigated intervals are already clearly highlighted in Figure 4C and we do not think it necessary to add them again to Figure 4B. We will add data pointer to the grain-size record in Figure 4C to better visualize the existing gaps in the record. 4.) Figure 5: Red noise level will be added to Figure 5A and 5B.

Table 1.) In the revised version of the Table 1 we will list the full planktic foraminifera species names. The citation Raffi et al. (2006) will be properly cited in the reference list in the revised manuscript.

Referee 2

Major comments

1.) As stated under Lines 232 to 234 in our initial manuscript, the argument regarding the intensification AMOC is based on the findings by Bell et al. (2015) and not just the SST data published by Lawrence et al. (2009) and shown in our Figure 4A. Including

the SST record in Figure 4 is for orientation purposes of the reader as to the onset of the prosed "plateau" in the North Atlantic SST development. The arguments made in Bell et al. (2015) are based already on a wide spread analysis of SST records across the North Atlantic including Sites (ODP Sites 607, 1090, 1082 and 982). However, we agree that including the studies by Khélifi and Frank (2014) and Lisiecki (2014) will be beneficiary for our study as they highlight the lack of increased overturning circulation in the deep water opposed to the increased overturning postulated by Bell et al. (2015) in relation to the surface water trajectory. In our opinion this would strongly argue for the effect of MOW on the intermediate branch of overturning circulation, a scenario already highlighted in Bahr et al. (2015) for MIS 5. This argument will be newly added into the revised manuscript (under subsection 4.3) and complement the existing argument of the prevalence of MOW along intermediate water death within the North Atlantic by Loubere (1987) (see Lines 228 to 239 of our initial manuscript) which was already included in the initial manuscript.

2.) The $\delta$18O correction between both benthic species for the Early Pleistocene is y=1.06x-0.17 (R2= 0.80). Hence, the slope of the linear relationship is $\sim$1 and the y-intercept is minor considering the analytical error of the measurements which is $\pm$0.08‰. This suggests a comparable oxygen isotope fractionation between Planulina ariminensis and Cibicidoides ungerianus. A similar behaviour has been postulated for P. ariminensis and other Cibicidoides species (e.g. Marchitto et al. 2014). For $\delta$13C, the computed correction factor for both benthic species during the Early Pleistocene is y=0.13x+0.84 (R2=0.02). Following the suggestion of Referee 2, we have reanalysed our data for a possible climate driven bias. Firstly, the samples utilized for the analysis of the interspecies correction were not specifically chosen for their warm/cold climatic background but under the premise that both benthic species were present in sufficient numbers for stable isotope analysis. Hence, the suggested form of analysis leads to the exclusion of $\sim$ 25% (n=20) samples from the original data set corresponding to transional climate conditions. This exclusion changes the inter species correlation to y=-0.02x+0.83 (R2< 0.02). The correlation for only "warm" climatic conditions

(corresponding light $\delta$18O values) shifts the interspecies correlation to y=0.005x+0.83 (R2<0.02; n=26). Similar, the correlation for only "cold" climatic conditions (corresponding heavy $\delta$18O values) shifts the interspecies correlation to y=-0.04x+0.84 (R2<0.02; n=29). Hence, it becomes obvious that no climatic driven bias can be found. We argue instead that the high scatter might relate to the variability of C. ungerianus from a preferably epifaunal to a very shallow infaunal life style in correspondence to different nutrient fluxes, oxygenation state, habitat changes etc. This would cause an enhanced variability in the $\delta$13C microhabitat-offset between both species. Such variability has been observed at recent for other Cibicidoides species (Fontanier et al., 2006). In contrast, P. ariminensis has been argued to be a reliable recorder of the $\delta$13C signal of MOW (Zahn et al., 1987; already stated on Lines 83 to 85 in the initial manuscript) and aligns with findings of e.g., Schönfeld (2002), Rogerson et al. (2011) and García-Gallardo et al. (2017) further suggesting that P. ariminensis is a true "elevated" epifaunal living species directly recording MOW properties. Specifically, the influence of remineralisation of sedimentary carbon on benthic $\delta$13C which may overprint the MOW signal was discussed by Rogerson et al. (2011). The authors considered the $\delta$13C signal ambiguous for most benthic foraminifera with the exception of P. ariminensis which showed the highest (positive) correlation with MOW flow strength. We will add the above stated information to section 2.3 (section 2.2 in the initial manuscript). This will further strengthen our argument to exclude the C. ungerianus $\delta$13C data points from the discussion and presentation in section 4.1 and Figure 4A. The reference in the caption of Figure 4 to Kaboth et al., in prep. is a typo and will be removed in the revised version of the manuscript. Only the stated inter species relations were applied in this study.

3.) We agree with the Referee 2 and will include the findings of Becker et al. (2006, 2005) on precession influence on climate variability during MIS 100 in the Mediterranean Sea into the revised manuscript under subsection 4.2.

4.) Vertical movements of the MOW plume are an important mechanism as stated by

Referee 2. However, the Late Pleistocene study of Bahr et al. (2015) has shown that Site U1389 is generally less prone to vertical movement than sites further up the shelf even under much more severe sea level variability than during the Early Pleistocene. The validity of utilizing $\delta$18O to trace MOW prevalence in the Gulf of Cadiz has been already established in Kaboth et al. (2016). This approach argues that MOW is the dominant water mass at the site with the heaviest oxygen isotopic signal compared to ambient North Atlantic water due to its density. We will highlight this statement by revising Figure 1 of our manuscript and add the modern vertical water mass distribution along T, S and $\delta$18Ow profiles for Site U1389 as Figure 1C. Based on this assumption the isotopic differences between the $\delta$18O of the Mediterranean Sea (input signal) and the Gulf of Cadiz (output signal) reflects MOW variability as the ice volume contribution for the same time interval in both stable oxygen isotope records can be assume to be identical. As the isotopic gradient in $\delta$18O are generally small throughout both intervals it seems feasible to argue that MOW prevailed throughout our studied time frame. The grain-size and $\delta$13C gradient for both intervals give indication that the intensification of MOW occurred as outlined in subsection 4.1 of the initial manuscript.

Minor comments

1.) We will follow the suggestion by Referee 2 and change the title into: "Mediterranean Outflow variability during the Early Pleistocene"

2.) Line 51: Following the suggestions by Referee 2 we will add more details on sequence stratigraphy and paleo-water depth of the Singa/Vrica sections in a newly designed subsection 2.2 under Material & Methods. It seems more befitting to add additional information for the reader about the Singa/Vrica section separately rather than into the Introduction.

3.) Line 112-118: The high variability at Site U1389 in sedimentation rate is not unusual if compared to findings from the same site during the Late Pleistocene which shows a similar range (see Figure DR2 in Bahr et al. 2015). Generally, contourites are very

dynamic depositional systems which is reflected in the evolution of sedimentation rates though time (Hernandez-Molina et al., 2014).

4.) Line 216: Yes, the black starts in the initial version of Figure 4B indicate the occurrences of N. atlantica during cold periods in Interval I of our study. As also suggested by Referee 1 (see above) we will modify Figure 4B to improve the visual occurrence pattern of N. atlantica to the reader also in relation to interval II.

5.) Line 263: We do not make this argument based on our data but instead this is based on the findings of Bell et al. (2015) as clearly stated in Lines 232 to 234.

6.) The superscription on $\delta$18O and $\delta$13C will be checked in the revised manuscript.

7.) Figure 1: This figure will be improved by adding a recent vertical water mass profile at Site U1389 including T, S, and $\delta$18Ow.

8.) Figure 2: The different climatic intervals will be highlighted, and it will be made clear to readers which figure represents stable oxygen and carbon correlations.

9.) Figure 3: Commas will be replaced by dots.

10.) Figure 4: As also suggested by Referee 1 we will improve Figure 4A by adding the South Atlantic SST record ODP 1090 to highlight the discussed intensification of AMOC (Bell et al. 2015). Furthermore, we will clarify the occurrence of N. atlantica (also see Referee 2 comments on Line 216 and Referee 1), and also visually improve the time range of the studied intervals in relation to the shown SST records.

References

Bahr, A., Kaboth, S., Jiménez-Espejo, F.J., Sierro, F.J., Voelker, A.H.L., Lourens, L., Röhl, U., Reichart, G.J., Escutia, C., Hernández-Molina, F.J., Pross, J., andFriedrich, O., 2015, Persistent monsoonal forcing of Mediterranean Outflow Water dynamics during the late Pleistocene: Geology, v. 43, p. 951–954, doi: 10.1130/G37013.1.

Becker, J., Lourens, L.J., Hilgen, F.J., van derLaan, E., Kouwenhoven, T.J., an-

dReichart, G.-J., 2005, Late Pliocene climate variability on Milankovitch to millennial time scales: A high-resolution study of MIS100 from the Mediterranean: Palaeogeography, Palaeoclimatology, Palaeoecology, v. 228, p. 338–360, doi: 10.1016/j.palaeo.2005.06.020.

Becker, J., Lourens, L.J., andRaymo, M.E., 2006, High-frequency climate linkages between the North Atlantic and the Mediterranean during marine oxygen isotope stage 100 (MIS100): ATLANTIC MEDITERRANEAN LINKAGE: Paleoceanography, v. 21, p. PA3002, doi: 10.1029/2005PA001168.

Bell, D.B., Jung, S.J.A., andKroon, D., 2015, The Plio-Pleistocene development of Atlantic deep-water circulation and its influence on climate trends: Quaternary Science Reviews, v. 123, p. 265–282, doi: 10.1016/j.quascirev.2015.06.026.

Fontanier, C., Mackensen, A., Jorissen, F.J., Anschutz, P., Licari, L., andGriveaud, C., 2006, Stable oxygen and carbon isotopes of live benthic foraminifera from the Bay of Biscay: Microhabitat impact and seasonal variability: Marine Micropaleontology, v. 58, p. 159–183, doi: 10.1016/j.marmicro.2005.09.004.

García-Gallardo, Á., Grunert, P., Van derSchee, M., Sierro, F.J., Jiménez-Espejo, F.J., Alvarez Zarikian, C.A., andPiller, W.E., 2017, Benthic foraminifera-based reconstruction of the first Mediterranean-Atlantic exchange in the early Pliocene Gulf of Cadiz: Palaeogeography, Palaeoclimatology, Palaeoecology, v. 472, p. 93–107, doi: 10.1016/j.palaeo.2017.02.009.

Hernandez-Molina, F.J., Stow, D.A.V., Alvarez-Zarikian, C.A., Acton, G., Bahr, A., Balestra, B., Ducassou, E., Flood, R., Flores, J.-A., Furota, S., Grunert, P., Hodell, D., Jimenez-Espejo, F., Kim, J.K., et al., 2014, Onset of Mediterranean outflow into the North Atlantic: Science, v. 344, p. 1244–1250, doi: 10.1126/science.1251306.

Kaboth, S., Lourens, L., andDepartement Aardwetenschappen (Utrecht), 2016, Deciphering the paleoceanographic and paleoclimatic evolution of the Gulf of Cádiz during

the past 2.6 million years. Khélifi, N., andFrank, M., 2014, A major change in North Atlantic deep water circulation 1.6 million years ago: Climate of the Past, v. 10, p. 1441–1451, doi: 10.5194/cp-10-1441-2014.

Lawrence, K.T., Herbert, T.D., Brown, C.M., Raymo, M.E., andHaywood, A.M., 2009, High-amplitude variations in North Atlantic sea surface temperature during the early Pliocene warm period: VARIABLE PLIOCENE NORTH ATLANTIC SSTS: Paleoceanography, v. 24, p. n/a-n/a, doi: 10.1029/2008PA001669. Lisiecki, L.E., 2014, Atlantic overturning responses to obliquity and precession over the last 3 Myr: Paleoceanography, v. 29, p. 71–86, doi: 10.1002/2013PA002505.

Lisiecki, L.E., andRaymo, M.E., 2005, A Pliocene-Pleistocene stack of 57 globally distributed benthic d18O records: Paleoceanography, v. 20, p. 1–17, doi: 10.1029/2004PA001071.

Loubere, P., 1987, Changes in mid-depth North Atlantic and Mediterranean circulation during the Late Pliocene — Isotopic and sedimentological evidence: Marine Geology, v. 77, p. 15–38, doi: 10.1016/0025-3227(87)90081-8.

Marchitto, T.M., Curry, W.B., Lynch-Stieglitz, J., Bryan, S.P., Cobb, K.M., andLund, D.C., 2014, Improved oxygen isotope temperature calibrations for cosmopolitan benthic foraminifera: Geochimica et Cosmochimica Acta, v. 130, p. 1–11, doi: 10.1016/j.gca.2013.12.034.

Raffi, I., Backman, J., Fornaciari, E., Pälike, H., Rio, D., Lourens, L., andHilgen, F., 2006, A review of calcareous nannofossil astrobiochronology encompassing the past 25 million years☆: Quaternary Science Reviews, v. 25, p. 3113–3137, doi: 10.1016/j.quascirev.2006.07.007.

Rogerson, M., Schönfeld, J., andLeng, M.J., 2011, Qualitative and quantitative approaches in palaeohydrography: A case study from core-top parameters in the Gulf of Cadiz: Marine Geology, v. 280, p. 150–167, doi: 10.1016/j.margeo.2010.12.008.

Schönfeld, J., 2002, A new benthic foraminiferal proxy for near-bottom current velocities in the Gulf of Cadiz, northeastern Atlantic Ocean: Deep-Sea Research Part I: Oceanographic Research Papers, v. 49, p. 1853–1875, doi: 10.1016/S0967-0637(02)00088-2.

Zahn, R., Sarnthein, M., andErlenkeuser, H., 1987, Benthic isotope evidence for changes of the Mediterranean outflow during the Late Quaternary: Paleoceanography, v. 2, p. 543–559, doi: 10.1029/PA002i006p00543.
* * *

---

## Author Response (AR1)

Authors:     Stefanie Kaboth, Patrick Grunert and Lucas J. Lourens

Title:     Mediterranean outflow Water variability during the Early Pleistocene

Journal:     Climate of the Past

Corresponding author's E-Mail address: stefaniekaboth@ntu.edu.tw

Dear Editor,

Firstly, we would like to thank our two anonymous Referees for their careful handling of the manuscript as well as for the constructive comments and suggestions which we kindly acknowledge.

The main criticism of the two referees generally concentrated on: (1) the validity of the grain-size distribution and its corresponding spectral analysis (Referee 1), (2) the applied benthic stable isotope correlation (Referee 2), and (3) the presentation of the data in Figure 4 (Referee 1 +2).

In the following, we would like to give a detailed explanation concerning the comments and recommendations given by the Reviewers, as well as the intended revisions in our revised manuscript.

Stefanie Kaboth, on behalf of all co-authors

**Referee 1**

*1.) Chapter Spectral Analysis & Precession control on MOW strength during the Early Pleistocene*

As requested by Referee 1, we have improved our spectral analysis by including wavelet analysis for both investigated intervals. These new results supplement the findings of the power spectra by clearly showing the dominance and stability of the (semi-)precession signal in the grain-size variability at Site U1389 during the Early Pleistocene. We have emphasized this finding accordingly in section 4.2 (Line 300 to 303) of our revised manuscript. The wavelets for both investigated intervals have been added to Figure 5 as Figures 5C and 5D. These adaptions are also in line with suggestions made by Referee 2 (see below), and we have now also included the findings of Becker et al. (2006, 2005) about precession related signals in the Eastern Mediterranean Sea during interval I (see Line 305 to 308). The methodological description of the wavelet analysis has been added to the Method section 2.7 (Line 185 to 187).

*2.) Chapter Glacial-Interglacial change*

We believe that the MIS102 interval is robust in its present version and no further adaptions are necessary. Following the age constrain of MIS 102 according to Lisiecki and Raymo, (2005) the interval extends from 2.575 to 2.554 Ma. Through the initial age model based on bio- and magnetostratigraphy (see Table 1) the age interval between 2.5 and 2.581 Ma is very well constrained.

In contrast to Referee 1, we argue that the grain-size variability is a reliable representation for MOW variability during our investigated intervals despite in-part low recovery and sample resolution. As stated in the manuscript (see Lines 157 to 163) the grain-size proxy has been successfully used for Late Pleistocene studies (i.e. Kaboth et al. 2017, 2016). We think that the argued absolute amplitude reduction in MOW flow strength between interval I and II (see Lines 280 to 282) is also visible in Figure 4D and does not rely solely on the calculated relative change in amplitude expressed in Line 280.

We also believe that the description of the change in $\delta^{13}C$ through North Atlantic water influence is sufficient in its current state (see Line 274 to 280).

*3.) Chapter: Did Mow contribute to the Early Pleistocene climate transition?*

The occurrence of *Neogloboquadrina atlantica* (sin) is now clearly denoted in the revised Figure 4A by arrows at mid-point level of-MIS 100, 98, 96 at Site U1389. The reference for abundance patterns of *N. atlantica* in the eastern Mediterranean Sea and North Atlantic are already provided in the discussion section 4.3 (Lines 379). The south Atlantic SST record of Site ODP 1090 has been added to Figure 4A following Bell et al. (2015) (this also aligns with suggestions made by Referee 2, see below).

*Figures*

1.) Figure 1C highlighting the Mediterranean Sea circulation has been added. Caption changes have been made accordingly.

2.) Figure 3: Indeed, the correlation arrow for MIS 101 in Figure 3A has been revised. The dotted line indicating the mean sedimentation rate in Figure 3C has been removed.

3.) In the now revised Figure 4 we have unifed the age notations between Figure 4 and Table 1. The mid-point levels of sapropels throughout the investigated intervals are already clearly highlighted in Figure 4D and we do not think it necessary to add them again to Figure 4B. We have added data pointer to the grain-size record in Figure 4D to better visualize the existing gaps in the record.

4.) Figure 5: Red noise level has been added to Figure 5A and 5B. Caption changes have been made accordingly.

*Table*

1.) In the now revised version of the Table 1 we have listed the full species names of the biostratigraphic markers. The citation Raffi et al. (2006) has been added in the reference list.

**Referee 2**

*Major comments*

1.) As stated under Lines 232 to 234 in our initial manuscript, the argument regarding the intensification AMOC is based on the findings by Bell et al. (2015) and not just the SST data published by Lawrence et al. (2009) and shown in our Figure 4A. Including the SST record in Figure 4 is for orientation purposes of the reader as to the onset of the prosed "plateau" in the North Atlantic SST development. The arguments made in Bell et al. (2015) are based already on a wide spread analysis of SST records across the North Atlantic including Sites (ODP Sites 607, 1090, 1082 and 982). However, we agree with Referee 2 and included the studies by Khélifi and Frank (2014) and Lisiecki (2014) (Line 365 to 372). These authors highlighted the lack of increased overturning circulation in the deep water opposed to the increased overturning postulated by Bell et al. (2015) in relation to the surface water trajectory. In our opinion this strongly argues for the effect of MOW on the intermediate branch of overturning circulation, a scenario already highlighted in Bahr et al. (2015) for MIS 5. This argument will has been newly added into the revised manuscript (under subsection 4.3, Line 365 to 372) and complement the existing argument of the prevalence of MOW along intermediate water death within the North Atlantic by Loubere (1987) which was already included in the initial manuscript.

2.) The $\delta^{18}O$ correction between both benthic species for the Early Pleistocene is y=1.06x-0.17 ($R^2$= 0.80). Hence, the slope of the linear relationship is ~1 and the y-intercept is minor considering the analytical error of the measurements which is ±0.08‰. This suggests a comparable oxygen isotope fractionation between *Planulina ariminensis* and *Cibicidoides ungerianus*. A similar behaviour has been postulated for *P. ariminensis* and other *Cibicidoides* species (e.g. Marchitto et al. 2014). For $\delta^{13}C$, the computed correction factor for both benthic species during the Early Pleistocene is y=0.13x+0.84 ($R^2$=0.02). Following the suggestion of Referee 2, we have reanalysed our data for a possible climate driven bias. Firstly, the samples utilized for the analysis of the interspecies correction were not specifically chosen for their warm/cold climatic background but under the premise that both benthic species were present in sufficient numbers for stable isotope analysis. Hence, the suggested form of analysis leads to the exclusion of ~ 25% (n=20) samples from the original data set corresponding to transional climate conditions. This exclusion changes the inter species correlation to y=-0.02x+0.83 ($R^2$< 0.02). The correlation for only "warm" climatic conditions (corresponding light $\delta^{18}$O values) shifts the interspecies correlation to y=0.005x+0.83 ($R^2$<0.02; n=26). Similar, the correlation for only "cold" climatic conditions (corresponding heavy $\delta^{18}$O values) shifts the interspecies correlation to y=-0.04x+0.84 ($R^2$<0.02; n=29). Hence, it becomes obvious that no climatic driven bias can be found. We argue instead that the high scatter might relate to the variability of *C. ungerianus* from a preferably epifaunal to a very shallow infaunal life style in correspondence to different nutrient fluxes, oxygenation state, habitat changes etc. This would cause an enhanced variability in the $\delta^{13}$C microhabitat-offset between both species. Such variability has been observed at recent for other *Cibicidoides* species (Fontanier et al., 2006). In contrast, *P. ariminensis* has been argued to be a reliable recorder of the $\delta^{13}$C signal of MOW (Zahn et al., 1987; already stated on Lines 83 to 85 in the initial manuscript) and aligns with findings of e.g., Schönfeld (2002), Rogerson et al. (2011) and García-Gallardo et al. (2017) further suggesting that *P. ariminensis* is a true "elevated" epifaunal living species directly recording MOW properties. Specifically, the influence of remineralisation of sedimentary carbon on benthic $\delta^{13}$C which may overprint the MOW signal was discussed by Rogerson et al. (2011). The authors considered the $\delta^{13}$C signal ambiguous for most benthic foraminifera with the exception of *P. ariminensis* which showed the highest (positive) correlation with MOW flow strength.

We have added the additional information regarding the $\delta^{18}$O and $\delta^{13}$C signals at Site U1389 to section 2.3 of our revised manuscript with the exception of the climatic bias as simply no evidence for this could be found (as outlined above). The reference in the caption of Figure 4 to Kaboth et al., *in prep.* was a typo and has been removed. Only the stated inter species relations were applied in this study.

3.) We agree with the Referee 2 and have now included the findings of Becker et al. (2006, 2005) on precession influence on climate variability during MIS 100 in the Mediterranean Sea into the revised manuscript under subsection 4.2 (Line 305 to 308).

4.) Vertical movements of the MOW plume are an important mechanism as stated by Referee 2. However, the Late Pleistocene study of Bahr et al. (2015) has shown that Site U1389 is generally less prone to vertical movement than sites further up the shelf even under much more severe sea level variability than during the Early Pleistocene. The validity of utilizing $\delta^{18}$O to trace MOW prevalence in the Gulf of Cadiz has been already established in Kaboth et al. (2016). This approach argues that MOW is the dominant water mass at the site with the heaviest oxygen isotopic signal compared to ambient North Atlantic water due to its density. We have highlighted this statement by revising Figure 1 of our manuscript and added the modern vertical water mass distribution along T, S and $\delta^{18}O_w$ profiles for Site U1389 as Figure 1D. Based on this assumption the isotopic differences between the $\delta^{18}O$ of the Mediterranean Sea (input signal) and the Gulf of Cadiz (output signal) reflects MOW variability as the ice volume contribution for the same time interval in both stable oxygen isotope records can be assume to be identical. As the isotopic gradient in $\delta^{18}O$ are generally small throughout both intervals it seems feasible to argue that MOW prevailed throughout our studied time frame. We have now outlined this approach in more detail in the revised manuscript in Line 245 to 257. The grain-size and $\delta^{13}C$ gradient for both intervals give indication that the intensification of MOW occurred as outlined in subsection 4.1 of the manuscript. No further additions have been made.

*Minor comments*

1.) We have followed the suggestion by Referee 2 and changed the title into: "Mediterranean Outflow variability during the Early Pleistocene"

2.) Following the suggestions by Referee 2 we have added more details on sequence stratigraphy and paleo-water depth of the Singa/Vrica sections in a newly designed subsection 2.2 under Material & Methods (Line 105 to 114). It seemed more befitting to add additional information for the reader about the Singa/Vrica section in the Material section rather than into the Introduction as suggested by the Referee.

3.) Line 112-118 (initial manuscript): The high variability at Site U1389 in sedimentation rate is not unusual if compared to findings from the same site during the Late Pleistocene which shows a similar range (see Figure DR2 in Bahr et al. 2015). Generally, contourites are very dynamic depositional systems which is reflected in the evolution of sedimentation rates though time (Hernandez-Molina et al., 2014). No further changes have been made to the revised manuscript.

4.) Line 216 (initial manuscript): Yes, the black starts in the initial version of Figure 4B indicated the occurrences of *N. atlantica* during cold periods in Interval I. In accordance to the suggestions made also by Referee 1 (see above) we have modified Figure 4B to improve the visual occurrence pattern of *N. atlantica* to the reader.

5.) Line 263 (initial manuscript): We do not make this argument based on our data but instead this is based on the findings of Bell et al. (2015) as clearly stated in Line 363 to 365.

6.) The superscription of $\delta^{18}O$ and $\delta^{13}C$ has been checked in the revised manuscript.

7.) We have added to Figure 1: the location map of Singa/Vrica sections under 1B, a schematic of the Mediterranean circulation under 1C, and the modern vertical water mass profile at Site U1389 including T, S, and $\delta^{18}Ow$ under 1D.

8.) We have highlighted in Figure 2A and B which represents stable oxygen and carbon isotope correlations.

9.) Figure 3: Commas have been replaced by dots.

10.) We believe Figure 4 has been visually improved to better reflect the content of the manuscript and aid the reader. As also suggested by Referee 1 we have added the South Atlantic SST record ODP 1090 to highlight the discussed intensification of AMOC (Bell et al. 2015). Furthermore, we have clarified the occurrence of *N. atlantica* (also see Referee 2 comments on Line 216 and Referee 1), and also visually improved the time range of the studied intervals in relation to the shown SST records.

**2.2 Singa and Vrica**

The Monte Singa IV and Vrica sections of Early Pleistocene age contain sequences of marine marls and sapropelic clay layers, which are exposed in Calabria, southern Italy (Lourens et al., 1992). During the time of deposition, both sections have been part of the continental slope bordering the Ionian basin. The benthic foraminiferal associations represent a deep bathyal paleoenvironment between ~900 to ~1100 m water depth (Verhallen, 1991). This suggests that the benthic isotope data derived from these sediment sequences recorded intermediate water mass conditions within the eastern Mediterranean Sea. The biostratigraphic correlation indicates that the Vrica sapropelite suite is equivalent to the IV sequence at Monte Singa (Verhallen, 1991; Zijderveld et al., 1991).

**2.3 Stable isotope measurements and interspecies correction**

The freeze-dried sediment samples of Site U1389 were wet sieved into three fractions (>150 μm, 150-63μm, 63-38 μm), and their residues oven dried at 40°C. Stable oxygen ($\delta^{18}$O) and carbon ($\delta^{13}$C) isotope analyses were carried out on 4 to 6 specimens of the epifaunal living foraminiferal species *Planulina ariminensis* and *Cibicidoides ungerianus* from the >150 μm size fraction. All selected specimens were crushed, sonicated in ethanol, and dried at 35˚C. Stable isotope analyses were carried out on a CARBO-KIEL automated carbonate preparation device linked to a Thermo-Finnigan MAT253 mass spectrometer at Utrecht University. The precision of the measurements is ±0.08‰ for $\delta^{18}$O and ±0.03 for $\delta^{13}$C. The results were calibrated using the international standard NBS-19, and the in-house standard NAXOS.

Isotopic values are reported in standard delta notation (δ) relative to the Vienna Pee Dee
Belemnite (VPDB). *P. ariminensis* was absent in 100 samples; resulting gaps were filled with
*C. ungerianus* values corrected for interspecies isotopic offsets. The calculation of the
interspecies offset is based on 62 paired isotope measurements of both benthic species. The
$\delta^{18}$O interspecies offset was determined by applying a least square linear regression equation
(Fig. 2). The Pearson correlation coefficient ($R^2$) between both species shows high correlation
of 0.80 for $\delta^{18}$O (Fig. 2A). The calculated slope of this relationship is ~1 with an y-intercept
of +0.10 ‰ which is minor considering the analytical error of the measurements of ±0.08‰.
This suggests a comparable oxygen isotope fractionation between *P. ariminensis* and *C.*
*ungerianus*. A similar behaviour has been postulated for *P. ariminensis* and other *Cibicidoides*
species (Marchitto et al., 2014). These results also align with findings from the same benthic
species during the Late and Middle Pleistocene (Kaboth et al., 2017). In contrast, the $\delta^{13}$C
correlation factor for both benthic species during the Early Pleistocene is insignificant $R^2$=0.02
(Fig. 2B). We argue that the high scatter of *C. ungerianus* during the Early Pleistocene might
relate to the variability from a preferably epifaunal to a very shallow infaunal life style in
correspondence to different nutrient fluxes, oxygenation state, habitat changes etc. This would
cause an enhanced variability in the $\delta^{13}$C microhabitat-offset between both species. Such
variability has been observed at recent for other *Cibicidoides* species (Fontanier et al., 2006).
In contrast, *P. ariminensis* has been argued to be a reliable recorder of the $\delta^{13}$C signal of MOW
(Zahn et al., 1987). Rogerson et al. (2011), Schönfeld (2002) and García-Gallardo et al. (2017)
further suggesting that *P. ariminensis* is a true "elevated" epifaunal living species directly
recording MOW properties. Specifically, the influence of remineralisation of sedimentary
carbon on benthic $\delta^{13}$C which may overprint the MOW signal was discussed by Rogerson et
al. (2011). The authors considered the $\delta^{13}$C signal ambiguous for most benthic foraminifera
with the exception of *P. ariminensis* which showed the highest (positive) correlation with
MOW flow strength. Therefore, we only present the $\delta^{13}$C of *P. ariminensis*, considered a
valuable basis for $\delta^{13}$C studies of the paleo-hydrography of the MOW.

2.4 Grain-size analyses

The stable isotope sample preparation was used to obtain weight percentages (wt.-%) of the
grain-size fractions >150 μm, 150-63μm, 63-38 μm and <38 μm for the investigated samples
were obtained during sample preparation for isotope analyses. We concentrate on the grain- size fraction between 63-150 µm which has been used previously as indicator for flow strength changes in the Gulf of Cadiz attributed to MOW variability (Rogerson et al., 2005). Even though untreated weight percentages hold a bias it has been shown for the last climatic cycle that weight percentages mirror major peaks in Zr/Al records, considered a reliable recorder of

MOW flow strength variability(Bahr et al., 2014), and thus can be used to trace MOW intensity patterns (Kaboth et al., 2016, 2017).

**2.5 Chronology**

[revised manuscript text omitted]

**3.3 Grain-size**

The mean grain-size values (63-150 µm) for both investigated intervals are ~8.0 %-wt. Highest values of both investigated intervals of up to ~60 %-wt. are correlated with MIS 100 and 77 (Fig. 4). The grain-size variability is seemingly not related to glacial-interglacial variability as a clear response of the grain-size to the variability of $\delta^{18}O$ records at Site U1389 cannot be observed.

**3.4 Spectral analyses**

The grain-size records of Interval I and II at Site U1389 exhibit significance (80% to 90%) variance in the precession (~23 kyr), semi-precession (~ 11 kyr) and potentially 1/3-precession (~7 kyr; significant Interval II only) frequency band (Fig. 5A and B). The obliquity signal is insignificant in both investigated intervals. The wavelet analysis for Interval I (Fig. 5C) reveals that the precession and semi-precession signal is most dominant between 2.55 and 2.50 Myrs. The lack of stability in the precession band from 2.5 to 2.4 Ma correlates with the reduced sample resolution due to poor core recovery (see Fig. 3). During Interval II the precession and semi-precession signal is most dominant during the interval between 2.0 Ma and 1.9 Ma. Starting from 2.0 Ma the 1/3-precession signal is becoming increasingly more prominent and stable (Fig. 5D).

**4. Discussion**

**4.1 Glacial-Interglacial MOW variability at Site U1389 during the Early Pleistocene**

In order to utilize the $\delta^{18}O$ signal at Site U1389 to trace MOW variability we assume that the global ice volume contributions of the $\delta^{18}O$ signal within the same time interval for Site U1389 and the Mediterranean Sea are equal. Consequently, differences in $\delta^{18}O$ are caused by temperature and/or salinity differences of the water masses between both sites. The modern heavy oxygen isotope signal of MOW (see Fig. 1D) is a consequence of its increased temperature and salinity linked to its Mediterranean source region, and hence setting it apart from the isotopic lighter overflowing water masses of North Atlantic origin. Therefore, we argue that the similarities of the $\delta^{18}$O values between Site U1389 and the MedSea stack during Interval I (2.6-2.4 Ma) and Interval II (2.1-1.8 Ma) emphasizes the direct influence of MOW at Site U1389. In this sense, our findings also strongly suggest that MOW formation during the Early Pleistocene was similar to modern conditions where MOW originates largely from intermediate water masses such as the Levantine Intermediate Water (Millot, 2009, 2014; Millot et al., 2006). The $\delta^{18}$O difference between Site U1389 and the Mediterranean Sea is small during glacial periods in both investigated intervals, suggesting that Site U1389 bathed in MOW during these colder climatic conditions throughout the Early Pleistocene time interval (Fig. 4A). This is particularly interesting in light of the proposed vertical shift of the MOW flow path during glacial periods of the Late Pleistocene fostered by the increased density of the outflowing Mediterranean water masses (Kaboth et al., 2016; Lofi et al., 2015; Rogerson et al., 2005; Schönfeld andZahn, 2000; Toucanne et al., 2007; Voelker et al., 2006). This suggests that Site U1389 was not subjected to major glacial-interglacial induced flow path changes during the early Pleistocene, possibly due to its deeper and relatively proximal location to the Strait of Gibraltar, placing it more into the general flow path of upper MOW. These results confirm the inferences derived from Site U1389 of the Late Pleistocene interval where MOW activity was also shown to be largely unaffected by glacial-interglacial variability but instead predominately influenced by insolation driven hydro-climatic changes of its Mediterranean source region (Bahr et al., 2015).

In contrast, the interglacial periods of both intervals show a small but relative depletion in the Mediterranean Sea compared to the $\delta^{18}$O signal at Site U1389 which might reflect relatively higher temperatures or lower salinity of the intermediate Mediterranean Sea waters with respect to the MOWs during interglacial periods. The strongest intervals of relative $\delta^{18}$O depletion throughout both investigated time periods correlate with MIS 103, 102, MIS 75 and MIS 67 characterized by a depletion of up to ~0.5 ‰ in the Mediterranean Sea compared to Site U1389. This shift might correspond to a freshening of the Mediterranean Sea intermediate water column during sapropel formation and a consequently reduction of MOW influence at Site U1389 (Rogerson et al., 2012). In case of MIS 102 and 67 sapropels have been documented in the Eastern Mediterranean Sea basin but not for MIS 75 (Emeis et al., 2000; Lourens, 2004; Lourens et al., 1992, 1996a). During Interval II, the generally heavier $\delta^{13}$C values at U1389 are close to those of the Mediterranean Sea values inferring that MOW was in fact the predominant source of bottom water at Site U1389 between 1.8 and 2.1 Ma (Fig. 4C). In contrast, the older

Interval I is characterized by a slightly increased $\delta^{13}C$ gradient between Site U1389 and the

Mediterranean Sea suggesting a generally larger contribution of ambient North Atlantic water masses carrying a lighter $\delta^{13}C$ signal to the site. This could indicate a more vigorous MOW or that during Interval I the MOW flow core was less proximal than during Interval II. The later argument seems to be supported by the grain-size and its variability, as Interval II shows a

~10% decrease in mean and amplitude relative to Interval I (Fig. 4D). This would suggest that during Interval I Site U1389 was less proximal to the flow core albeit more sensitive to flow strength changes whereas during Interval II the MOW plume has settled upon Site U1389. This is further supported by findings from seismic records in the Gulf of Cadiz that also suggest that at ~2.1 Ma the present day circulation established (Hernandez-Molina et al., 2014b).

A distinct increase in the $\delta^{13}C$ gradient can be seen during MIS 96, which may document a particular strong MOW activity. However, the sample resolution during MIS 96

and the subsequent MIS 95 is relatively low so that increase in the $\delta^{13}C$ gradient remains ambiguous. The onset of the subsequent hiatus which has been argued to represent depositional erosion due to increased bottom current activity of the MOW could argue for a strong intensification of MOW activity (Hernandez-Molina et al., 2014b).

4.2 Precession control on MOW strength during the Early Pleistocene: Similarities to Late

Pleistocene MOW behaviour?

Untreated grain-size weight percentages can only give an indication for patterns in flow strength (Kaboth et al., 2016, 2017). For the two investigated intervals we find that the 63-150

µm fraction variability is seemingly modulated by a ~23 kyr pacing (Fig. 4D). This relationship is evident in the power spectrum of the grain-size data which yields for both intervals a dominance in the precession and semi-precession frequency band (~23 and ~11 kyr) (Fig. 5A

and B). The dominance and stability of the recorded precessional and semi-precessional signal in the grain-size variability throughout both investigated intervals is also highlighted by the wavelet analysis (Fig. 5C and D). This suggests that the flow strength of MOW was probably directly modulated by precession during the Early Pleistocene, aligning with previous findings based on Zr/Al ratios at Site U1389 from the Late Pleistocene (Bahr et al., 2015). In fact, a strong processional influence was also shown for $\delta^{18}O$ records from the eastern Mediterranean

Sea (ODP site 967 and 969) and the mid-latitude North Atlantic during MIS 100 to MIS 96

(Becker et al., 2005, 2006). For the late Pleistocene, an inverse relationship was found between precession and MOW dynamics (Bahr et al., 2015; Kaboth et al., 2016). During periods of increased summer insolation at the time of precession minima, the monsoonal rain belts expand northward causing an increase of freshwater discharge by the river Nile (e.g. Rohling et al., 2015; Rossignol-Strick, 1983, 1985). This effectively impedes intermediate water mass formation in the Eastern Mediterranean, thereby suppressing MOW production. From the correlation of the filtered ~23 kyr signal to the grain-size variability at site U1389 a similar relationship already existed during both investigated intervals of the Early Pleistocene (Fig. 4D). We also find significant semi -precession (~11 kyr) influence indicative for a primarily low-latitude response argued to originate in the tropics (Rutherford andD'Hondt, 2000; deWinter et al., 2014).

[revised manuscript text omitted]

1997.

Becker, J., Lourens, L. J., Hilgen, F. J., van derLaan, E., Kouwenhoven, T. J. and Reichart, G.-

J.: Late Pliocene climate variability on Milankovitch to millennial time scales: A high- resolution study of MIS100 from the Mediterranean, Palaeogeogr. Palaeoclimatol. Palaeoecol.,

228(3–4), 338–360, doi:10.1016/j.palaeo.2005.06.020, 2005.

Becker, J., Lourens, L. J. and Raymo, M. E.: High-frequency climate linkages between the

North Atlantic and the Mediterranean during marine oxygen isotope stage 100 (MIS100),

Paleoceanography, 21(3), PA3002, doi:10.1029/2005PA001168, 2006.

Bell, D. B., Jung, S. J. A. and Kroon, D.: The Plio-Pleistocene development of Atlantic deep- water circulation and its influence on climate trends, Quat. Sci. Rev., 123, 265–282, doi:10.1016/j.quascirev.2015.06.026, 2015.

Borenäs, K. M., Wåhlin, A. K., Ambar, I. and Serra, N.: The Mediterranean outflow splitting—

a comparison between theoretical models and CANIGO data, Deep Sea Res. Part II Top. Stud.

Oceanogr., 49, 4195–4205, doi:10.1016/S0967-0645(02)00150-9, 2002.

Bryden, H. L. and Stommel, H. M.: Liminting processes thatdetermine basic features of the circulation in the Mediterranean Sea, Oceanol. Acta, 7(3), 289–296, 1984.

Bryden, H. L., Candela, J. and Kinder, T. H.: Exchange through the Strait of Gibraltar, Prog.

Oceanogr., 33, 201–248, doi:10.1016/0079-6611(94)90028-0, 1994.

Cramp, A. and O'Sullivan, G.: Neogene sapropels in the Mediterranean: a review, Mar. Geol.,

153(1–4), 11–28, doi:10.1016/S0025-3227(98)00092-9, 1999.

Emeis, K.-C., Sakamoto, T., Wehausen, R. and Brumsack, H.-J.: The sapropel record of the eastern Mediterranean Sea — results of Ocean Drilling Program Leg 160, Palaeogeogr.

Palaeoclimatol. Palaeoecol., 158(3–4), 371–395, doi:10.1016/S0031-0182(00)00059-6, 2000.

Etourneau, J., Schneider, R., Blanz, T. and Martinez, P.: Intensification of the Walker and

Hadley atmospheric circulations during the Pliocene–Pleistocene climate transition, Earth

Planet. Sci. Lett., 297(1–2), 103–110, doi:10.1016/j.epsl.2010.06.010, 2010.

Fiúza, A. F. G., Hamann, M., Ambar, I., Díaz del Río, G., González, N. and Cabanas, J. M.:

Water masses and their circulation off western Iberia during May 1993, Deep Sea Res. Part I

Oceanogr. Res. Pap., 45(7), 1127–1160, doi:10.1016/S0967-0637(98)00008-9, 1998.

García-Gallardo, Á., Grunert, P., Van derSchee, M., Sierro, F. J., Jiménez-Espejo, F. J.,
Alvarez Zarikian, C. A. and Piller, W. E.: Benthic foraminifera-based reconstruction of the first
Mediterranean-Atlantic exchange in the early Pliocene Gulf of Cadiz, Palaeogeogr.
Palaeoclimatol. Palaeoecol., 472, 93–107, doi:10.1016/j.palaeo.2017.02.009, 2017.

Gouhier, T. C., Grinstead, A. and Simko, V.: biwavelet: Conduct univariate and bivariate
wavelet    analyses    (Version    0.20.10).    [online]    Available    from:
http://github.com/tgouhier/biwavelet, 2016.

Gradstein, F. M., Ogg, J. G. and Hilgen, F. J.: On The Geologic Time Scale, Newsletters
Stratigr., 45(2), 171–188, doi:10.1127/0078-0421/2012/0020, 2012.

Grinsted, A., Moore, J. C. and Jevrejeva, S.: Application of the cross wavelet transform and
wavelet coherence to geophysical time series, Nonlinear Process. Geophys., 11(5/6), 561–566,
doi:10.5194/npg-11-561-2004, 2004.

Grunert, P., Balestra, B., Richter, C., Flores, J.-A., Auer, G., Garcia Gallardo, A. and Piller, W.
E.: Revised and refined age model  for the upper Pliocene of IODP Site U1389 (IODP Exp.
339, Gulf of Cadiz), Newsl. Stratigr., doi:10.1127/nos/2017/0396, 2017.

Hernandez-Molina, F. J., Llave, E., Preu, B., Ercilla, G., Fontan, A., Bruno, M., Serra, N.,
Gomiz, J. J., Brackenridge, R. E., Sierro, F. J., Stow, D. A.V., Garcia, M., Juan, C., Sandoval,
N. and Arnaiz, A.: Contourite processes associated with the Mediterranean Outflow Water after
its exit from the Strait of Gibraltar: Global and conceptual implications, Geology, 42(3), 227–
230, doi:10.1130/G35083.1, 2014a.

Hernandez-Molina, F. J., Stow, D. A.V., Alvarez-Zarikian, C. A., Acton, G., Bahr, A., Balestra,
B., Ducassou, E., Flood, R., Flores, J.-A., Furota, S., Grunert, P., Hodell, D., Jimenez-Espejo,
F., Kim, J. K., Krissek, L., Kuroda, J., Li, B., Llave, E., Lofi, J., Lourens, L., Miller, M.,
Nanayama, F., Nishida, N., Richter, C., Roque, C., Pereira, H., Sanchez Goni, M. F., Sierro, F.
J., Singh, A. D., Sloss, C., Takashimizu, Y., Tzanova, A., Voelker, A., Williams, T. and Xuan,
C.: Onset of Mediterranean outflow into the North Atlantic, Science (80-. )., 344(6189), 1244–
1250, doi:10.1126/science.1251306, 2014b.

Hernández-Molina, F. J., Llave, E., Stow, D. A. V., García, M., Somoza, L., Vázquez, J. T.,
Lobo, F. J., Maestro, A., Díaz del Río, V., León, R., Medialdea, T. and Gardner, J.: The
contourite depositional system of the Gulf of Cádiz: A sedimentary model related to the bottom
current activity of the Mediterranean outflow water and its interaction with the continental margin, Deep Sea Res. Part II Top. Stud. Oceanogr., 53(11–13), 1420–1463, doi:10.1016/j.dsr2.2006.04.016, 2006.

Hernández-Molina, F. J., Stow, D., Alvarez-Zarikian, C., Acton, G., Bahr, A., Balestra, B., Ducassou, E., Flood, R., Flores, J. A., Furota, S., Grunert, P., Hodell, D., Jimenez-Espejo, F., Kim, J. K., Krissek, L., Kuroda, J., Li, B., Llave, E., Lofi, J., Lourens, L., Miller, M., Nanayama, F., Nishida, N., Richter, C., Roque, C., Pereira, H., Goñi Fernanda Sanchez, M., Sierro, F. J., Singh, A. D., Sloss, C., Takashimizu, Y., Tzanova, A., Voelker, A., Williams, T. and Xuan, C.: IODP Expedition 339 in the Gulf of Cadiz and off West Iberia: Decoding the environmental significance of the Mediterranean outflow water and its global influence, Sci. Drill., 16, 1–11, doi:10.5194/sd-16-1-2013, 2013.

Hernández-Molina, F. J., Sierro, F. J., Llave, E., Roque, C., Stow, D. A.V, Williams, T., Lofi, J., Van derSchee, M., Arnaiz, A., Ledesma, S., Rosales, C., Rodriguez-Tovar, F. J., Pardo-Iguzquiza, E. andBrackenridge, R. E.: Evolution of the Gulf of Cadiz margin and southwest Portugal contourite depositional system: Tectonic, sedimentary and paleoceanographic implications from IODP expedition 339, Mar. Geol., 377, 7–39, doi:10.1016/j.margeo.2015.09.013, 2015.

Kaboth, S., Bahr, A., Reichart, G.-J., Jacobs, B. and Lourens, L. J.: New insights into upper MOW variability over the last 150kyr from IODP 339 Site U1386 in the Gulf of Cadiz, Mar. Geol., 377, 136–145, doi:10.1016/j.margeo.2015.08.014, 2016.

Kaboth, S., deBoer, B., Bahr, A., Zeeden, C. and Lourens, L. J.: Mediterranean Outflow Water dynamics during the past ~570 kyr: Regional and Global implications: Mid- to Late Pleistocene MOW, Paleoceanography, doi:10.1002/2016PA003063, 2017.

Khelifi, N., Sarnthein, M., Andersen, N., Blanz, T., Frank, M., Garbe-Schonberg, D., Haley, B. A., Stumpf, R. and Weinelt, M.: A major and long-term Pliocene intensification of the Mediterranean outflow, 3.5-3.3 Ma ago, Geology, 37(9), 811–814, doi:10.1130/G30058A.1, 2009.

Khélifi, N. andFrank, M.: A major change in North Atlantic deep water circulation 1.6 million years ago, Clim. Past, 10(4), 1441–1451, doi:10.5194/cp-10-1441-2014, 2014.

Khélifi, N., Sarnthein, M., Frank, M., Andersen, N. and Garbe-Schönberg, D.: Late Pliocene variations of the Mediterranean outflow, Mar. Geol., 357, 182–194, doi:10.1016/j.margeo.2014.07.006, 2014.

Lang, D. C., Bailey, I., Wilson, P. A., Chalk, T. B., Foster, G. L. and Gutjahr, M.: Incursions
of southern-sourced water into the deep North Atlantic during late Pliocene
glacial?intensification, Nat. Geosci., 9(5), 375–379, doi:10.1038/ngeo2688, 2016.

Lawrence, K. T., Herbert, T. D., Brown, C. M., Raymo, M. E. and Haywood, A. M.: High-
amplitude variations in North Atlantic sea surface temperature during the early Pliocene warm
period, Paleoceanography, 24(2), PA2218, doi:10.1029/2008PA001669, 2009.

Lisiecki, L. E.: Atlantic overturning responses to obliquity and precession over the last 3 Myr,
Paleoceanography, 29(2), 71–86, doi:10.1002/2013PA002505, 2014.

Liu, Y., San Liang, X. and Weisberg, R. H.: Rectification of the Bias in the Wavelet Power
Spectrum, J. Atmos. Ocean. Technol., 24(12), 2093–2102, doi:10.1175/2007JTECHO511.1,
2007.

Llave, E., Schönfeld, J., Hernández-Molina, F. J., Mulder, T., Somoza, L., Díaz Del Río, V.
and Sánchez-Almazo, I.: High-resolution stratigraphy of the Mediterranean outflow contourite
system in the Gulf of Cadiz during the late Pleistocene: The impact of Heinrich events, Mar.
Geol., 227, 241–262, doi:10.1016/j.margeo.2005.11.015, 2006.

Lofi, J., Voelker, A. H. L., Ducassou, E., Hernández-Molina, F. J., Sierro, F. J., Bahr, A.,
Galvani, A., Lourens, L. J., Pardo-Igúzquiza, E., Pezard, P., Rodríguez-Tovar, F. J. and
Williams, T.: Quaternary chronostratigraphic framework and sedimentary processes for the
Gulf of Cadiz and Portuguese Contourite Depositional Systems derived from Natural Gamma
Ray records, Mar. Geol., doi:10.1016/j.margeo.2015.12.005, 2015.

Loubere, P.: Changes in mid-depth North Atlantic and Mediterranean circulation during the
Late Pliocene — Isotopic and sedimentological evidence, Mar. Geol., 77(1–2), 15–38,
doi:10.1016/0025-3227(87)90081-8, 1987.

Lourens, L. J.: Revised tuning of Ocean Drilling Program Site 964 and KC01B (Mediterranean)
and implications for the δ 18 O, tephra, calcareous nannofossil, and geomagnetic reversal
chronologies of the past 1.1 Myr, Paleoceanography, 19(3), PA3010,
doi:10.1029/2003PA000997, 2004.

Lourens, L. J.: On the Neogene-Quaternary debate, Episodes, 31, 239–242, 2008.

Lourens, L. J. and Hilgen, F. J.: Long-periodic variations in the earth's obliquity and their
relation to third-order eustatic cycles and late Neogene glaciations, Quat. Int., 40, 43–52, doi:10.1016/S1040-6182(96)00060-2, 1997.

Lourens, L. J., Hilgen, F. J., Gudjonsson, L. and Zachariasse, W. J.: Late Pliocene to early
Pleistocene astronomically forced sea surface productivity and temperature variations in the
Mediterranean, Mar. Micropaleontol., 19(1–2), 49–78, doi:10.1016/0377-8398(92)90021-B,
1992.

Lourens, L. J., Hilgen, F. J., Raffi, I. and Vergnaud-Grazzini, C.: Early Pleistocene chronology
of the Vrica Section (Calabria, Italy), Paleoceanography, 11(6), 797–812,
doi:10.1029/96PA02691, 1996a.

Lourens, L. J., Antonarakou, A., Hilgen, F. J., VanHoof, A. A. M., Vergnaud-Grazzini, C. and
Zachariasse, W. J.: Evaluation of the Plio-Pleistocene astronomical timescale,
Paleoceanography, 11(4), 391–413, doi:10.1029/96PA01125, 1996b.

Marchitto, T. M., Curry, W. B., Lynch-Stieglitz, J., Bryan, S. P., Cobb, K. M. and Lund, D. C.:
Improved oxygen isotope temperature calibrations for cosmopolitan benthic foraminifera,
Geochim. Cosmochim. Acta, 130, 1–11, doi:10.1016/j.gca.2013.12.034, 2014.

Martinez-Garcia, A., Rosell-Mele, A., McClymont, E. L., Gersonde, R. and Haug, G. H.:
Subpolar Link to the Emergence of the Modern Equatorial Pacific Cold Tongue, Science
(80-. )., 328(5985), 1550–1553, doi:10.1126/science.1184480, 2010.

Millot, C.: Another description of the Mediterranean Sea outflow, Prog. Oceanogr., 82, 101–
124, doi:10.1016/j.pocean.2009.04.016, 2009.

Millot, C.: Heterogeneities of in- and out-flows in the Mediterranean Sea, Prog. Oceanogr.,
120, 254–278, doi:10.1016/j.pocean.2013.09.007, 2014.

Millot, C., Candela, J., Fuda, J.-L. and Tber, Y.: Large warming and salinification of the
Mediterranean outflow due to changes in its composition, Deep Sea Res. Part I Oceanogr. Res.
Pap., 53(4), 656–666, doi:10.1016/j.dsr.2005.12.017, 2006.

Mulder, T., Lecroart, P., Hanquiez, V., Marches, E., Gonthier, E., Guedes, J.-C., Thiébot, E.,
Jaaidi, B., Kenyon, N., Voisset, M., Perez, C., Sayago, M., Fuchey, Y. and Bujan, S.: The
western part of the Gulf of Cadiz: contour currents and turbidity currents interactions, Geo-
Marine Lett., 26(1), 31–41, doi:10.1007/s00367-005-0013-z, 2006.

Myers, P. G.: Flux-forced simulations of the paleocirculation of the Mediterranean,
Paleoceanography, 17(1), 1009, doi:10.1029/2000PA000613, 2002.

Patterson, M. O., McKay, R., Naish, T., Escutia, C., Jimenez-Espejo, F. J., Raymo, M. E.,

Meyers, S. R., Tauxe, L., Brinkhuis, H., Klaus, A., Fehr, A., Bendle, J. A. P., Bijl, P. K., Bohaty,

S. M., Carr, S. A., Dunbar, R. B., Flores, J. A., Gonzalez, J. J., Hayden, T. G., Iwai, M., Katsuki,

K., Kong, G. S., Nakai, M., Olney, M. P., Passchier, S., Pekar, S. F., Pross, J., Riesselman, C.

R., Röhl, U., Sakai, T., Shrivastava, P. K., Stickley, C. E., Sugasaki, S., Tuo, S., van deFlierdt,

T., Welsh, K., Williams, T. and Yamane, M.: Orbital forcing of the East Antarctic ice sheet during the Pliocene and Early Pleistocene, Nat. Geosci., 7(11), 841–847, doi:10.1038/ngeo2273, 2014.

Peliz, A., Marchesiello, P., Santos, A. M. P., Dubert, J., Teles-Machado, A., Marta-Almeida,

M. and LeCann, B.: Surface circulation in the Gulf of Cadiz: 2. Inflow-outflow coupling and the Gulf of Cadiz slope current, J. Geophys. Res., 114(C3), doi:10.1029/2008JC004771, 2009.

Peliz, Á., Dubert, J., Santos, A. M. P., Oliveira, P. B. and LeCann, B.: Winter upper ocean circulation in the Western Iberian Basin—Fronts, Eddies and Poleward Flows: an overview,

Deep Sea Res. Part I Oceanogr. Res. Pap., 52(4), 621–646, doi:10.1016/j.dsr.2004.11.005,

2005.

R Core Team: R: A language and environment for statistical computing, [online] Available from: http://www.r-project.org/, 2014.

Raffi, I., Backman, J., Fornaciari, E., Pälike, H., Rio, D., Lourens, L. and Hilgen, F.: A review of calcareous nannofossil astrobiochronology encompassing the past 25 million years☆, Quat.

Sci. Rev., 25(23–24), 3113–3137, doi:10.1016/j.quascirev.2006.07.007, 2006.

Raymo, M. E., Hodell, D. and Jansen, E.: Response of deep ocean circulation to initiation of northern hemisphere glaciation (3-2 MA), Paleoceanography, 7(5), 645–672, doi:10.1029/92PA01609, 1992.

Rogerson, M., Rohling, E. J., Weaver, P. P. E. and Murray, J. W.: Glacial to interglacial changes in the settling depth of the Mediterranean Outflow plume, Paleoceanography, 20(3),

1–12, doi:10.1029/2004PA001106, 2005.

Rogerson, M., Rohling, E. J. and Weaver, P. P. E.: Promotion of meridional overturning by

Mediterranean-derived salt during the last deglaciation, Paleoceanography, 21(4), PA4101, doi:10.1029/2006PA001306, 2006.

Rogerson, M., Schönfeld, J. and Leng, M. J.: Qualitative and quantitative approaches in palaeohydrography: A case study from core-top parameters in the Gulf of Cadiz, Mar. Geol.,
280(1–4), 150–167, doi:10.1016/j.margeo.2010.12.008, 2011.

Rohling, E. J., Marino, G. andGrant, K. M.: Mediterranean climate and oceanography, and the
periodic development of anoxic events (sapropels), Earth-Science Rev., 143, 62–97,
doi:10.1016/j.earscirev.2015.01.008, 2015.

Rossignol-Strick, M.: African monsoons, an immediate climate response to orbital insolation,
Nature, 304(5921), 46–49, doi:10.1038/304046a0, 1983.

Rossignol-Strick, M.: Mediterranean Quaternary sapropels, an immediate response of the
African monsoon to variation of insolation, Palaeogeogr. Palaeoclimatol. Palaeoecol., 49(3–4),
237–263, doi:10.1016/0031-0182(85)90056-2, 1985.

Rutherford, S. and D'Hondt, S.: Early onset and tropical forcing of 100,000-year Pleistocene
glacial cycles, Nature, 408(6808), 72–75, doi:10.1038/35040533, 2000.

Schönfeld, J.: A new benthic foraminiferal proxy for near-bottom current velocities in the Gulf
of Cadiz, northeastern Atlantic Ocean, Deep. Res. Part I Oceanogr. Res. Pap., 49, 1853–1875,
doi:10.1016/S0967-0637(02)00088-2, 2002.

Schönfeld, J. and Zahn, R.: Late Glacial to Holocene history of the Mediterranean outflow.
Evidence from benthic foraminiferal assemblages and stable isotopes at the Portuguese margin,
Palaeogeogr. Palaeoclimatol. Palaeoecol., 159, 85–111, doi:10.1016/S0031-0182(00)00035-3,
2000.

Schulz, M. and Mudelsee, M.: REDFIT: estimatingred-noise spectra directly from unevenly
spaced paleoclimatic time series, Comput. Geosci., 28(28), 421–426, 2002.

Shackleton, N. J. and Hall, M. A.: Oxygen and carbon isotope stratigraphy of the deep sea
drilling project hole 552A: Plio-Pleistocene glacial history, Initial Reports DSDP, 81, 599–609.
doi:10.1029/2000PA000513, 1984.

Stow, D. A.V., Hernández-Molina, F. J. and Alvarez-Zarikian, C.: Expedition 339 Summary,
edited by Expedtion 339 Scientists, Exped. 339 Summ., Proceeding(339),
doi:10.2204/iodp.proc.339.104.2013, 2013.

Torrence, C. and Compo, G. P.: A Practical Guide to Wavelet Analysis, Bull. Am. Meteorol.
Soc., 79(1), 61–78, doi:10.1175/1520-0477(1998)079<0061:APGTWA>2.0.CO;2, 1998.

Toucanne, S., Mulder, T., Schönfeld, J., Hanquiez, V., Gonthier, E., Duprat, J., Cremer, M.
and Zaragosi, S.: Contourites of the Gulf of Cadiz: A high-resolution record of the
paleocirculation of the Mediterranean outflow water during the last 50,000 years, Palaeogeogr.
Palaeoclimatol. Palaeoecol., 246, 354–366, doi:10.1016/j.palaeo.2006.10.007, 2007.

Verhallen, P. J.: Late Pliocene to Early Pleistocene Mediterranean mud-dwelling foraminifera:
influence of a changing environment on community structure and evolution., 1991.

Voelker, A., Lebreiro, S., Schonfeld, J., Cacho, I., Erlenkeuser, H. and Abrantes, F.:
Mediterranean outflow strengthening during northern hemisphere coolings: A salt source for
the glacial Atlantic?, Earth Planet. Sci. Lett., 245(1–2), 39–55, doi:10.1016/j.epsl.2006.03.014,
2006.

Voelker, A. H. L., Colman, A., Olack, G., Waniek, J. J. and Hodell, D.: Oxygen and hydrogen
isotope signatures of Northeast Atlantic water masses, Deep Sea Res. Part II Top. Stud.
Oceanogr., 116, 89–106, doi:10.1016/j.dsr2.2014.11.006, 2015.

Weaver, P.P.E andClement, B.M.: Magnetobiostratigraphy of planktonic foraminiferal datums:
Deep Sea Drilling Project Leg 94, North Atlantic, vol. 94, U.S. Government Printing Office.
1987.

deWinter, N. J., Zeeden, C. and Hilgen, F. J.: Low-latitude climate variability in the Heinrich
frequency band of the Late Cretaceous greenhouse world, Clim. Past, 10(3), 1001–1015,
doi:10.5194/cp-10-1001-2014, 2014.

Zachariasse, W. J., Gudjonsson, L., Hilgen, F. J., Langereis, C. G., Lourens, L. J., Verhallen,
P. J. J. M. and Zijderveld, J. D. A.: Late Gauss to Early Matuyama invasions of
Neogloboquadrina Atlantica in the Mediterranean and associated record of climatic change,
Paleoceanography, 5(2), 239–252, doi:10.1029/PA005i002p00239, 1990.

Zahn, R., Sarnthein, M. and Erlenkeuser, H.: Benthic isotope evidence for changes of the
Mediterranean outflow during the Late Quaternary, Paleoceanography, 2(6), 543–559,
doi:10.1029/PA002i006p00543, 1987.

Zijderveld, J. D. A., Hilgen, F. J., Langereis, C. G., Verhallen, P. and Zachariasse, W. J.:
Integrated magnetostratigraphy and biostratigraphy of the upper Pliocene-lower Pleistocene
from the Monte Singa and Crotone areas in Calabria, Italy, Earth Planet. Sci. Lett., 107(3–4),
697–714, 1991.

**Figure Captions**

Figure 1: (**A**) Study area with illustration of modern MOW pathways modified after (Bahr et al. 2015). Site location of U1389 (yellow dot) is marked. (**B**) Overview map of the Mediterranean Sea. Location of the Singa and Vrica sections in Italy (yellow dot) are marked. Black square indicates Gulf of Cadiz study area. (**C**) Water mass circulation in the Mediterranean Sea (modified after Cramp and O'Sullivan, 1999). MAW = Mixed Atlantic surface water; LIW = Levantine intermediate water; EMDW = Eastern Mediterranean deep water; WMDW = Western Mediterranean deep water (**D**) CTD depth profile of temperature (red line) and salinity (blue line) at Site U1389 derived from the World Ocean Database 2013. The data points of $\delta^{18}O_{water}$ (black line) are derived from neighbouring EUROFLEETS-Iberian-Forams Cruise site IB-F9 (36° 48.40' N; 7° 42.85'W) (Voelker et al., 2015). $NACW_{st}$=North Atlantic Central water of subtropical origin; $NACW_{sp}$= North Atlantic water of subpolar origin; MOW=Mediterranean Outflow water.

Figure 2: The $\delta^{18}O$ (**A**) and $\delta^{13}C$ (**B**) interspecies correlation between benthic foraminifera *Cibicidoides ungerianus* and *Planulina ariminensis* at Site U1389. Parallel measurements were conducted throughout both investigated intervals. Linear square regression (black line) equation and Pearson correlation coefficient ($R^2$) are shown.

Figure 3: Chronology of Site U1389. Assigned marine isotope stages (MIS) follow Lourens et al. (2004). (**A**) Both intervals of the $\delta^{18}O$ record of Site U1389 on shipboard mbsf scale correlated to the benthic $\delta^{18}O$ record of the Mediterranean Sea (MedSea stack) after Lourens et al. (1996a, unpublished data). Chronostratigraphy of MedSea stack is based on tuning sapropel midpoints to La2004 65º N summer insolation (Lourens, 2004). Lines with arrows indicate selected tie points used for the age model (a full list of tie points is available in Table 2). Black triangles with numbers indicating used biostratigraphic and paleomagnetic tie points as referenced in Table 1. Black and white bar at the top represents core recovery following Hernández-Molina et al. (2013) (**B**) Comparison of the benthic $\delta^{18}O$ record of Site U1389 on new time scale according to our tuning, and the benthic $\delta^{18}O$ MedSea stack on its respective age model (Lourens et al. 2004) (**C**) Calculated sedimentation rates for Site U1389.

Figure 4: (**A**) $UK^{37}$ based sea-surface temperature (SST) record of North Atlantic Site ODP 982 (Lawrence et al., 2009) and South Atlantic Site ODP 1090 (Martinez-Garcia et al., 2010). The running mean has a band width of 23. AMOC phases are marked by black arrows and follow the chronology of Bell et al. (2015). (**B**) Benthic $\delta^{18}O$ records of both investigated intervals at Site U1389. Interval I comprises the time frame of 2.6 to 2.4 Ma and Interval II 2.1

to 1.8 Ma. Isotopic gradient between both records is indicated by the grey-shaded area. (**C**)

Comparison of $\delta^{13}$C of *P. ariminensis* for both investigated intervals at Site U1389 and $\delta^{13}$C of the MedSea stack (Lourens et al. 1996a, unpublished data). The running means have a band width of 5. The *C. ungerianus* based $\delta^{13}$C values of the MedSea stack were adjusted to *P.*

*ariminensis* $\delta^{13}$C values of Site U1389 following the interspecies correction presented in this study (**D**) Grain-size (63-150 µm wt.-%) records for both investigated intervals at Site U1389.

The filtered ~23 kyr signal (f = 0.05±0.01) of the grain-size signal is indicated by the black dotted-line. Sapropel mid-points are marked by orange arrows and follow the chronology of

Emeis et al. (2000).

Figure 5: REDFIT power spectra of the grain-size values (63-150µm fraction in wt.-%) for both investigated intervals of Site U1389: (**A**) Interval I = 2.6-2.4 Ma and (**B**) Interval II: 2.1-

1.8 Ma). The 90% (red), 80% (blue) and AR1 red noise (black) confidence levels are given.

(**C**) Wavelet analysis of the grain-size values (63-150µm fraction in wt.-%) during Interval I

and (**D**) Wavelet analysis of the grain-size values (63-150µm fraction in wt.-%) during Interval

II. Cone of confidence (white) for both Intervals is marked. Areas with >95% significance level are marked by black lines. Periods corresponding to (semi, 1/3)-precession are marked with dashed white lines.

**Figure 1**

[Figure]

**Figure 2**

[Figure]

**Figure 3**

[Figure]

**Figure 4**

[Figure]

**Figure 5**

[Figure]

**Table Captions**

Table 1: Paleomagnetic and biostratigraphic tie points used in the primary age model of Site U1389 based on shipboard data following Hernández-Molina et al. (2013) and Stow et al. (2013). 1 = Gradstein et al. (2012); 2 = Raffi et al. (2006); 3= Lourens et al. (2004); 4 = (Grunert et al., 2017)

Table 2: Paleomagnetic and biostratigraphic tie points used in the primary age model of Site U1389 based on shipboard data following Hernández-Molina et al. (2013) and Stow et al. (2013). 1 = Gradstein et al. (2012); 2 = Raffi et al. (2006); 3= Lourens et al. (2004); 4 = Grunert et al. 2017

**Table 1**

| No. | Event | TOP Depth (mbsf) | BOT Depth (mbsf) | Age (ka) | Ref. |
|---|---|---|---|---|---|
| 1 | Top Olduvai | 542.00 | | 1806 | 1 |
| 2 | Bottom Olduvai | | 592.00 | 1945 | 1 |
| 3 | Matuyama/Gauss | 699.00 | | 2581 | 4 |
| 4 | LO *Calcidiscus* macintyrie | 510.99 | 515.65 | 1660 | 2 |
| 5 | FO *Globorotalia* inflata | 627.21 | 630.21 | 2090 | 3 |
| 6 | LO *Globorotalia* puncticulata | 645.02 | 646.61 | 2410 | 3 |
| 7 | LO *Discoaster* pentradiatus | 674.25 | 681.98 | 2500 | 2 |
| 8 | LO *Discoaster* scurlus | 681.98 | 693.70 | 2530 | 2 |
| 9 | LO *Discoaster* tamalis | 799.75 | 800 | 2800-2830 | 4 |

 **Table 2**

| Depth (mbsf) | Age (ka) |
|:---:|:---:|
| 512 | 1660 |
| 542 | 1806 |
| 551.25 | 1828 |
| 554 | 1851 |
| 564 | 1861 |
| 570 | 1867 |
| 574 | 1875 |
| 580 | 1898 |
| 592 | 1945 |
| 595 | 1965 |
| 600 | 1975 |
| 615.63 | 2005 |
| 623 | 2070 |
| 629.1 | 2092 |
| 629.75 | 2117.5 |
| 631.1 | 2132.5 |
| 646 | 2425 |
| 648.75 | 2435.5 |
| 665.1 | 2462.5 |
| 666.3 | 2486 |
| 673 | 2500 |
| 677.45 | 2517.5 |
| 687 | 2539 |
| 689 | 2552 |
| 691.5 | 2560 |
| 693.5 | 2583 |
| 699 | 2581 |
| 799.75 | 2800 |